# Retrograde fibroblast growth factor 22 (FGF22) signaling regulates insulin-like growth factor 2 (IGF2) expression for activity-dependent synapse stabilization in the mammalian brain

Akiko Terauchi[1], Erin M Johnson-Venkatesh[1†], Brenna Bullock[1†], Maria K Lehtinen[2], Hisashi Umemori[1*]

[1]Department of Neurology, F.M. Kirby Neurobiology Center, Boston Children's Hospital, Harvard Medical School, Boston, United States; [2]Department of Pathology, F.M. Kirby Neurobiology Center, Boston Children's Hospital, Harvard Medical School, Boston, United States

*For correspondence: hisashi.
umemori@childrens.harvard.edu

†These authors contributed
equally to this work

Competing interests: The
authors declare that no
competing interests exist.

Reviewing editor: Eunjoon Kim,
Korea Advanced Institute of
Science and Technology,
Republic of Korea

**Abstract** Communication between pre- and postsynaptic cells promotes the initial organization of synaptic specializations, but subsequent synaptic stabilization requires transcriptional regulation. Here we show that fibroblast growth factor 22 (FGF22), a target-derived presynaptic organizer in the mouse hippocampus, induces the expression of insulin-like growth factor 2 (IGF2) for the stabilization of presynaptic terminals. FGF22 is released from CA3 pyramidal neurons and organizes the differentiation of excitatory nerve terminals formed onto them. Local application of FGF22 on the axons of dentate granule cells (DGCs), which are presynaptic to CA3 pyramidal neurons, induces IGF2 in the DGCs. IGF2, in turn, localizes to DGC presynaptic terminals and stabilizes them in an activity-dependent manner. IGF2 application rescues presynaptic defects of $Fgf22^{-/-}$ cultures. IGF2 is dispensable for the initial presynaptic differentiation, but is required for the following presynaptic stabilization both in vitro and in vivo. These results reveal a novel feedback signal that is critical for the activity-dependent stabilization of presynaptic terminals in the mammalian hippocampus.

## Introduction

Synapses are the sites of neuronal communication in the brain. Proper synapse formation is critical for appropriate brain function; aberrant synaptic connectivity may result in various neurological and psychiatric disorders, such as autism, Fragile X syndrome, epilepsy, and schizophrenia (*Banerjee et al., 2014*; *Casillas-Espinosa et al., 2012*; *Lisman, 2012*; *Pfeiffer and Huber, 2009*). Synapse formation begins with target recognition by axons, which is followed by synaptic differentiation at the contact sites. Synaptic differentiation is regulated by signals that are exchanged between pre- and postsynaptic sites. Various target-derived presynaptic organizers, such as fibroblast growth factors (FGFs), WNTs, neurotrophins, neuroligins, Ephs/ephrins, SynCAMs, netrin-G ligands (NGLs), and signal regulatory proteins (SIRPs) are shown to promote local differentiation of presynaptic terminals (*Darabid et al., 2014*; *Fox and Umemori, 2006*; *Henriquez et al., 2011*; *Johnson-Venkatesh and Umemori, 2010*; *Regehr et al., 2009*; *Salinas, 2012*; *Shen and Scheiffele, 2010*; *Siddiqui and Craig, 2011*; *Toth et al., 2013*; *Zweifel et al., 2005*).

**eLife digest** Nerve cells in the developing brain must organize themselves into complex networks by forming appropriate connections with one another. These connections are known as synapses, and they assemble via two critical stages. First, a new synapse forms, and then it stabilizes. This first stage is a localized event that involves the contact site between the two nerve cells, while the stabilization of a synapse requires the expression of genes in a nerve cell's nucleus. Furthermore, only active synapses may be stabilized.

Many synapses form in a region of the brain called the hippocampus, which plays a key role in learning and memory. A protein called fibroblast growth factor 22 (or FGF22 for short) helps synapses to initially form within the hippocampus. However, much less is known about the signals that regulate the stabilization of synapses and the genes that are involved. It is also not clear if these genes might be controlled by FGF22 signaling.

To address these questions, Terauchi et al. searched the mouse hippocampus for genes with expression that depended on FGF22 signaling. One gene in particular, which encodes a protein called insulin-like growth factor 2 (IGF2), was much less expressed in mice that lack FGF22 compared to normal mice. Further experiments revealed that only active nerve cells transport IGF2 to synapses, and that IGF2 helps to stabilize these structures. By contrast, IGF2 is not required for synapse to initially form. This indicates that FGF22 controls both the formation and stabilization of synapses, and that it controls the first stage directly, and the second stage indirectly via its effects on IGF2 expression.

Terauchi et al. also showed that FGF22-IGF2 signaling is not involved in the stabilization of all synapses in the mouse hippocampus. Instead, synapses between different types of nerve cell appear to use distinct signals for synapse formation and stabilization. A key topic for future studies will be to understand these specific signals and how they cooperate in the brain to establish precise networks of nerve cells.

Initial synapses thereafter mature, resulting in a more stable, functional, and finely tuned neural network (*Goda and Davis, 2003*; *Waites et al., 2005*; *West and Greenberg, 2011*). Presynaptic stabilization has been shown to require gene expression. At the *Drosophila* larval neuromuscular junction (NMJ), a retrograde signal initiated by glass bottom boat (Gbb), the *Drosophila* homologue of bone morphogenic protein (BMP), controls presynaptic growth and stabilization through transcriptional regulation in motor neurons. During this process, Gbb and its receptor are internalized and transported from the nerve terminal to the cell body as a retrograde signal. This signal then activates a transcription factor, Mothers against decapentaplegic (Mad) (*Aberle et al., 2002*; *Marqués et al., 2002*; *McCabe et al., 2003*). Activated Mad regulates transcription of genes including *Trio* and *dfmr1* (fly homolog of FMR1). Trio and dFMR1 play critical roles in modulating actin cytoskeletal dynamics and stabilizing microtubules in the presynaptic motor neurons, leading to presynaptic growth and stabilization (*Ball et al., 2010*; *Nahm et al., 2013*). In the mammalian brain, changes in gene expression, as a consequence of axon–dendrite contacts, are also likely to contribute to presynaptic stabilization. For example, expression of genes encoding vesicle proteins increases soon after synaptogenesis begins, and neurons synthesize different isoforms of vesicle proteins before and after their axons contact targets (*Campagna et al., 1997*; *Lou and Bixby, 1995*; *Plunkett et al., 1998*; *Sanes and Lichtman, 1999*). In addition, synaptic stabilization is influenced by neural activity (*Ackermann et al., 2015*; *Chia et al., 2013*; *Dalva et al., 2007*; *Lichtman and Colman, 2000*; *Ruthazer and Cline, 2004*; *Waites et al., 2005*). However, it is not known whether and how target-derived molecules control gene transcription in the presynaptic neurons for the stabilization of presynaptic terminals, and whether such a pathway is regulated by neural activity.

We have previously found that FGF22 serves as a target-derived presynaptic organizer in the mouse hippocampus, a key brain region associated with learning, memory, emotional processing, and social behavior. FGF22 is released from CA3 pyramidal neurons and promotes local differentiation of the excitatory presynaptic terminals formed onto them (*Terauchi et al., 2010*; *2015*). FGF22-dependent presynaptic differentiation requires two FGF receptors (FGFRs), FGFR2b and FGFR1b, in

dentate granule cells (DGCs), the major presynaptic neurons for CA3 pyramidal neurons, and the downstream signaling molecules FGFR substrate 2 (FRS2) and PI-3 kinase (*Dabrowski et al., 2015*). Signals mediated by FRS2 and PI-3 kinase are known to regulate gene expression. Therefore, we hypothesized that FGF22 signaling eventually regulates gene expression and that those FGF22-induced molecules, in turn, contribute to the stabilization of presynaptic terminals.

Here, we identified FGF22 target genes in the presynaptic DGCs and asked whether the target genes contribute to presynaptic stabilization. We find that i) target-derived FGF22 signaling induces the expression of the insulin-like growth factor 2 (*Igf2*) gene in DGCs, ii) IGF2 then localizes to pre-synaptic terminals of DGCs and stabilizes them, iii) the transportation of IGF2 to the presynaptic terminal is activity-dependent, and iv) IGF2 is not required for the initial presynaptic differentiation, but is required for subsequent presynaptic stabilization both in vitro and in vivo. Thus, FGF22 is a target-derived molecule not only organizing local, initial presynaptic differentiation, but also regulating IGF2 expression in the presynaptic neurons. IGF2, in turn, contributes to presynaptic stabilization in an activity-dependent manner. Our results reveal a novel feedback signal that is critical for the activity-dependent stabilization of presynaptic terminals in the mammalian brain.

## Results

### Inactivation of FGF22 decreases IGF2 expression in DGCs during synapse formation

FGF signals are involved in the development of many organs via regulation of gene expression (*Chen et al., 2012*; *Mazzoni et al., 2013*). We hypothesized that in the hippocampus, the excitatory presynaptic organizer FGF22 would activate gene expression in the presynaptic neurons for the stabilization of presynaptic terminals. We focused on genes expressed in DGCs, because they provide a major excitatory input to CA3 pyramidal neurons, and their presynaptic differentiation is dependent on FGF22–FGFR signaling (*Dabrowski et al., 2015*). In the hippocampus, synapse formation starts in the first postnatal week and finishes by postnatal day 28 (P28) (*Danglot et al., 2006*; *Steward and Falk, 1991*). Presynaptic defects in $Fgf22^{-/-}$ mice begin to appear as early as P8 and are evident at P14 (*Terauchi et al., 2010*). To identify FGF22-regulated genes, we dissected P14 DGCs and compared gene expression profiles between wild-type (WT) and $Fgf22^{-/-}$ mice. Microarray analysis revealed several genes that are downregulated in $Fgf22^{-/-}$ DGCs relative to controls (*Table 1*). One of the most significantly downregulated genes was *Igf2*. This down-regulation was confirmed by RT-PCR (*Figure 1—figure supplement 1*), qPCR (*Igf2* was decreased to 53.1 ± 12.2% in $Fgf22^{-/-}$ mice relative to WT mice), and in situ hybridization. In situ hybridization experiments showed that at P14, and not at P7, expression of *Igf2* mRNA was decreased in DGCs (*Figure 1A–D*). It was not decreased in other hippocampal cells such as CA1 and CA3 pyramidal neurons of $Fgf22^{-/-}$ mice (*Figure 1A–D*). Interestingly, *Igf2* expression was clearly decreased in the inner layer of

**Table 1.** List of genes that are significantly downregulated in DGCs of $Fgf22^{-/-}$ mice at P14. Genes with Diff Score < -33 (p-value <0.001) relative to WT are shown in the table.

| Symbol | Definition | Diff Score |
|--------|------------|------------|
| *Slc6a13* | Solute carrier family 6 (Neurotransmitter Transporter), member 13 | -56.042 |
| *Col6a1* | Collagen, type VI, alpha 1 | -52.859 |
| *Prelp* | Proline arginine-rich end leucine-rich repeat | -50.844 |
| *Igf2* | Insulin-like growth factor 2 | -42.987 |
| *Mrc2* | Mannose receptor, C Type 2 | -41.904 |
| *Gys3* | Glycogen synthase 3, brain | -41.834 |
| *Aebp1* | Adipocyte enhancer-binding protein 1 | -40.069 |
| *Lrrtm3* | Leucine rich repeat transmembrane neuronal 3 | -35.357 |
| *Zfp365* | Zinc finger protein 365 | -34.604 |
| *Col1a1* | Collagen, type I, alpha 1 | -33.444 |

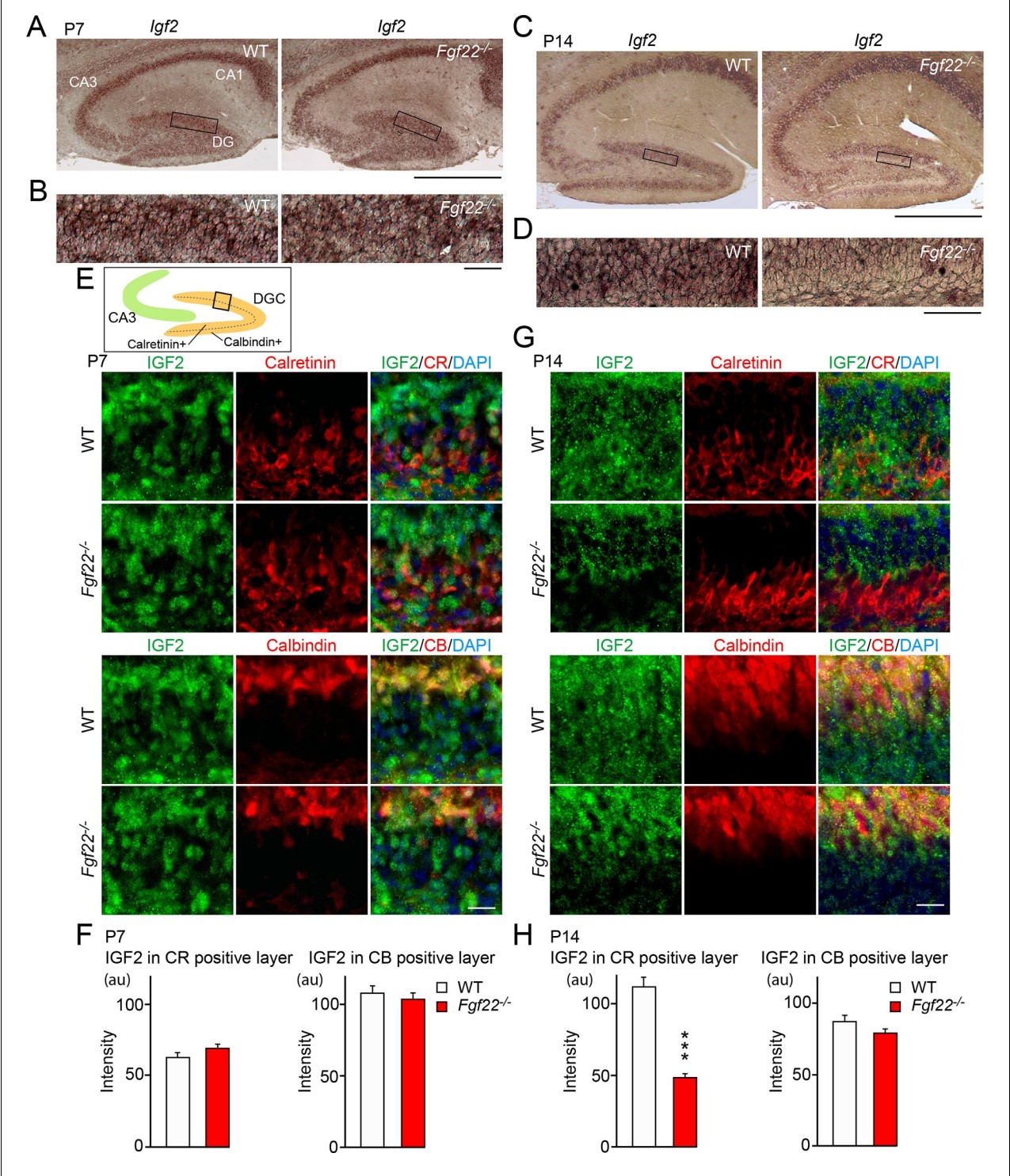

**Figure 1.** IGF2 expression is decreased in young DGCs in *Fgf22⁻/⁻* mice during the stage of synapse stabilization. (**A–D**) In situ hybridization for *Igf2* mRNA. (**A**) At P7 (initial stage of synaptic differentiation), *Igf2* mRNA is similarly expressed in the hippocampus of WT and *Fgf22⁻/⁻* mice. Higher magnification views of the boxed areas are shown in (**B**). (**C**) At P14 (around the time of synaptic stabilization), *Igf2* mRNA is decreased in *Fgf22⁻/⁻* mice in the inner molecular layer of DGCs. Higher magnification views of the boxed areas are shown in (**D**). Observations are from 3–5 animals per age and strain. (**E–H**) P7 and P14 hippocampal sections from WT and *Fgf22⁻/⁻* mice were immunostained for IGF2 and for either calretinin (CR; young DGCs) or calbindin (CB; mature DGCs). The illustration shows the pictured area (boxed). (**E**) IGF2 expression in DGCs at P7. Quantification of IGF2 immunoreactivity in CR and CB layers is shown in (**F**). There is no significant difference in the IGF2 intensity in either layer of DGCs at P7. (**G**) IGF2 expression in DGCs at P14. Quantification of IGF2 intensity in CR and CB layers is shown in (**H**). In P14 *Fgf22⁻/⁻* mice, IGF2 is significantly decreased in

*Figure 1 continued on next page*

*Figure 1 continued*

the CR-positive layer, but not in CB-positive layer of DGCs. Error bars are s.e.m. Data are from 77–160 fields from 3–5 animals. Significant difference from control at ***p<0.0001 by Student's t-test. Scale bars, (A and C) 500 µm, (B and D) 50 µm, (E and G) 20 µm.

The following figure supplements are available for figure 1:

**Figure supplement 1.** RT-PCR showing decreased *Igf2* mRNA expression in DGCs of P14 *Fgf22⁻/⁻* mice.

**Figure supplement 2.** Validation of the anti-IGF2 antibody.

*Fgf22⁻/⁻* DGCs, where relatively immature DGCs are located (*Figure 1C and D*) (*Aguilar-Arredondo et al., 2015*).

We then asked whether IGF2 expression was decreased in a specific developmental stage of DGCs in *Fgf22⁻/⁻* mice. Using specific markers, DGCs can be classified into several developmental subsets: from newborn to mature, DGCs are positive for Ki67 (dividing DGCs), doublecortin (immature), calretinin (young), and calbindin (mature) (*Abrous et al., 2005*). Ki67-, doublecortin-, and calretinin-positive DGCs populate the inner granule cell layer, and calbindin-positive DGCs the outer. We found that IGF2 protein expression was decreased in calretinin-positive, but not in calbindin-positive DGCs in *Fgf22⁻/⁻* mice relative to WT mice at P14 (*Figure 1G and H*; the specificity of the anti-IGF2 antibody was verified using tissues from *Igf2⁻/⁻* mice: *Figure 1—figure supplement 2*). Consistent with the in situ results, no changes were observed at P7 (*Figure 1E and F*). These results indicate that the lack of FGF22 impairs IGF2 expression in young, developing DGCs during the stage of synapse stabilization.

## Inactivation of FGF22 preferentially in CA3 pyramidal neurons decreases IGF2 expression in DGCs

FGF22 is highly expressed by CA3 pyramidal neurons in the hippocampus (*Terauchi et al., 2010*). We next examined whether FGF22 derived from CA3 pyramidal neurons is responsible for IGF2 expression in DGCs. For this, we inactivated FGF22 preferentially in CA3 pyramidal neurons using *Fgf22flox/flox* mice (*Fgf22f/f*; EUCOMM) crossed with *Grik4-Cre* mice (*Figure 2A*). *Grik4-Cre* mice express Cre in 100% of CA3 pyramidal neurons and 10% of DGCs in the hippocampus (*Nakazawa et al., 2002*). We found that at P14, IGF2 expression in the inner layer, but not in the outer layer, of DGCs of *Fgf22f/f::Grik4-Cre* mice was significantly decreased relative to that of control littermates (controls include wild type and *Fgf22f/f* mice; we did not observe any significant differences in IGF2 staining between wild type and *Fgf22f/f* mice) (*Figure 2B–D*). The level of decrease in IGF2 expression in the inner layer of *Fgf22f/f::Grik4-Cre* mice was similar to that in *Fgf22⁻/⁻* mice. These results suggest that CA3-derived FGF22 regulates the expression of IGF2 in young DGCs in vivo.

## Bath and axonal FGF22 application increases the expression of IGF2 in young DGCs

We next examined whether IGF2 expression in DGCs is increased by FGF22 treatment in culture. DGCs were identified with a marker, Prox1 (*Iwano et al., 2012*). In our hippocampal culture, initial formation of glutamatergic synapses starts from ∼3 days in vitro (3DIV), followed by activity-dependent synapse maturation from ∼8DIV to ∼12DIV (*Terauchi et al., 2010*; *Toth et al., 2013*). When cultured hippocampal cells were treated with FGF22 at 1DIV, IGF2 expression in the soma of DGCs was significantly increased at 7DIV (*Figure 3A and B*). We next asked whether only specific developmental subsets of DGCs increase IGF2 expression in response to FGF22 treatment. We found that FGF22-dependent increase in IGF2 was observed in calretinin-positive DGCs, but not in calbindin-positive DGCs (*Figure 3C and E*). In contrast, IGF2 expression did not increase in non-DGCs, such as CA3 pyramidal neurons, which were identified by immunostaining with Py-antibody (*Figure 3D and E*). These results demonstrate that FGF22 signaling regulates IGF2 expression in young, calretinin-positive DGCs. Note that at 7DIV, a majority of DGCs in culture were calretinin-positive (63.65 ± 1.40%; *Figure 3—figure supplement 1*), indicating that our results with DGCs (identified as Prox1-positive cells) mostly reflect calretinin-positive DGCs.

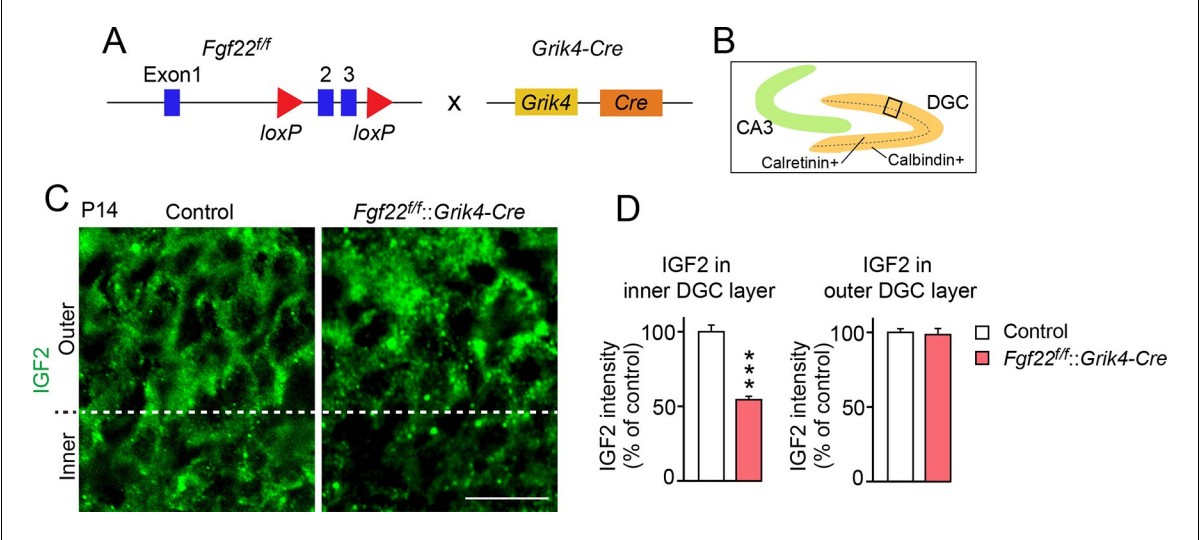

**Figure 2.** IGF2 expression in young DGCs is decreased in CA3-selective *Fgf22*-knockout mice. (**A**) Schematic of CA3-selective *Fgf22* deletion: *Fgf22*[flox/flox](*Fgf22*[f/f]) mice were crossed with mice carrying *Grik4*-promoter-driven Cre (*Grik4-Cre*). (**B–D**) IGF2 staining in the DGCs of P14 *Fgf22*[f/f]*::Grik4-Cre* mice and control littermates (WT and *Fgf22*[f/f] mice; we did not observe any significant differences in IGF2 staining between WT and *Fgf22*[f/f] mice). (**B**) Illustration showing the pictured area (boxed). (**C**) Representative pictures of IGF2 immunostaining in DGCs. The dashed line indicates the border between the inner and outer layers of DGCs. (**D**) Quantification of IGF2 immunoreactivity in the inner and outer DGC layers. CA3-selective inactivation of FGF22 results in a significant decrease in the IGF2 expression in the inner DGC layer, but not in the outer DGC layer. Error bars are s.e.m. Data are from 20–25 fields (**D**, inner DGC layer) and from 40–50 fields (**D**, outer DGC layer) from 4–5 animals. Significant difference from control at \*\*\*p<0.0001 by Student's t-test. Scale bar, 20 μm.

FGF22 is a target-derived presynaptic organizer that acts on axons. Hence, we next investigated whether local treatment with FGF22 at DGC axons is sufficient to increase expression of IGF2 in DGCs. Cultured hippocampal neurons were divided into axonal and somal compartments using an in vitro microfluidic culture system (*Figure 3F*). Axons of cultured hippocampal cells appeared in the axonal compartment by 2DIV. Local application of FGF22 at 2DIV to the axonal compartment resulted in an increase in IGF2 level in the cell bodies of Prox1-positive DGCs at 8DIV (*Figure 3G–I*), indicating that retrograde FGF22 signaling from axon terminals increases IGF2 expression in the soma of DGCs.

## IGF2 localizes to presynaptic terminals of DGCs and promotes their development

Application of FGF22 increases IGF2 expression in the DGC soma. We next asked where the induced IGF2 localizes in the DGCs. To address this question, we transfected EGFP-tagged IGF2 (IGF2-EGFP) and analyzed its localization in cultured DGCs. IGF2-EGFP showed a punctate pattern in neurofilament-positive axons of DGCs, while it was dim and diffuse throughout MAP2-positive dendrites of these neurons (*Figure 4A*). IGF2-EGFP puncta in the axons of cultured DGCs were colocalized with cotransfected synaptophysin-mCherry (83.1 ± 1.0% of IGF2-EGFP puncta were colocalized with synaptophysin-mCherry; *Figure 4B*), indicating that IGF2 localizes to presynaptic terminals. We then asked whether IGF2, which is a secreted protein, is localized on the surface of presynaptic terminals. We stained IGF2-EGFP transfected neurons with the anti-GFP antibody without a detergent followed by Alexa Fluor 647 secondary antibody (*Figure 4C*). We found that 41.2 ± 0.9% of IGF2-EGFP was localized on the cell surface. Surface IGF2-EGFP was always colocalized with synaptophysin-mCherry (*Figure 4C*), suggesting that IGF2 is secreted and tethered on the surface of presynaptic terminals. Next, we examined the localization of IGF2 receptors. IGF2R, the major receptor for IGF2 (*Fernandez and Torres-Aleman, 2012*), showed a punctate pattern in the axons of DGCs (*Figure 4—figure supplement 1A*) and was localized at presynaptic terminals (84.32 ± 0.90% of IGF2R puncta were colocalized with VGLUT1 puncta; *Figure 4—figure supplement 1B*). These

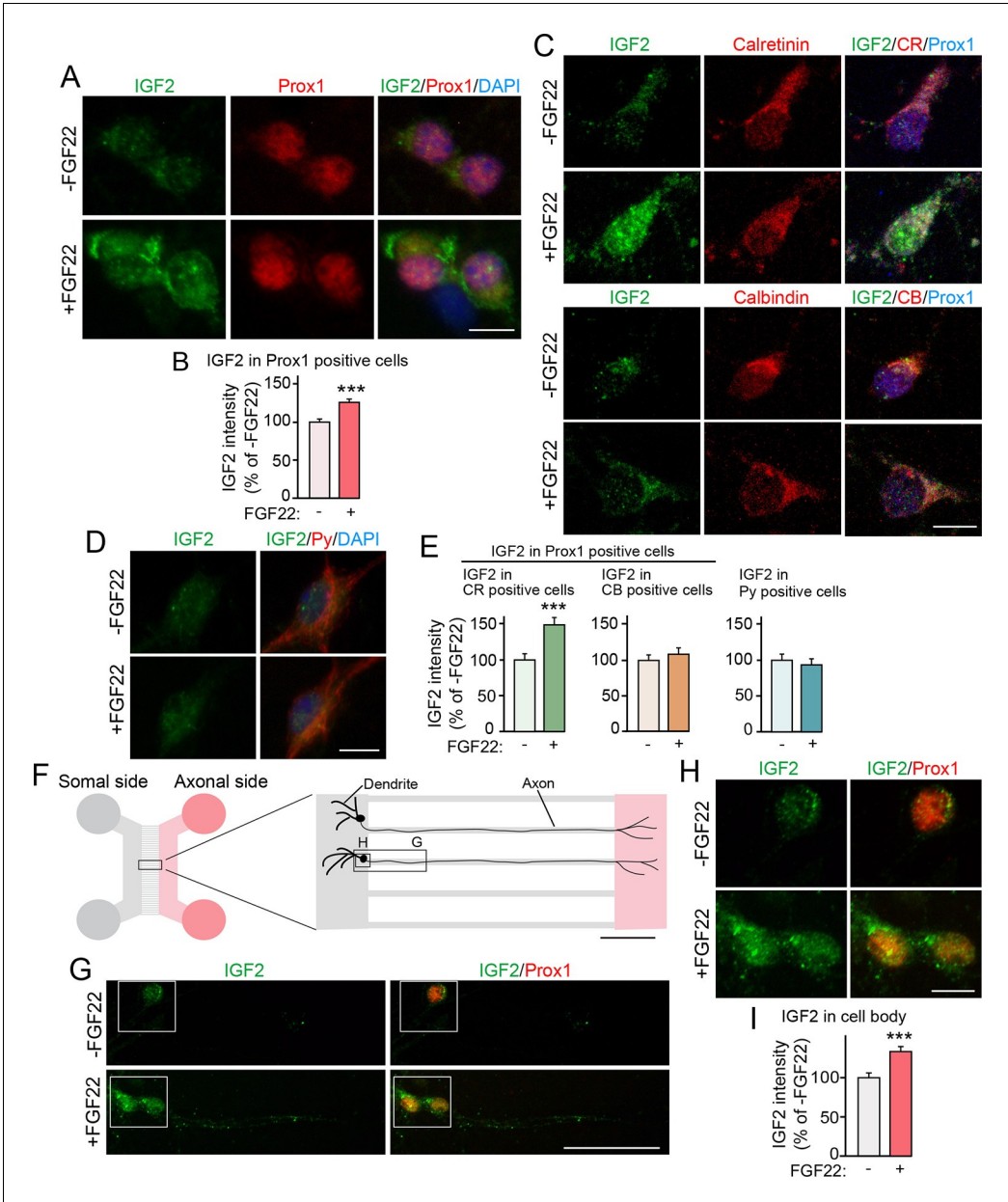

**Figure 3.** Bath and axonal application of FGF22 increases IGF2 expression in young DGCs. (**A–E**) Cultured hippocampal neurons were treated with FGF22 at 1DIV, and fixed and stained at 7DIV. (**A**) Bath application of FGF22 increases IGF2 expression in DGCs (Prox1-positive). (**B**) Quantification of IGF2 immunoreactivity in the cell bodies of DGCs, normalized to untreated condition. (**C**) Bath application of FGF22 increases IGF2 expression in CR-positive DGCs, but not in CB-positive DGCs. (**D**) Bath application of FGF22 does not affect IGF2 expression in CA3 pyramidal neurons (Py-positive). (**E**) Quantification of IGF2 immunoreactivity in the cell bodies of CR- or CB-positive DGCs and CA3 pyramidal neurons. Data are normalized to the intensity from untreated cells. (**F–I**) Hippocampal neurons were plated onto the somal compartment of microfluidic chambers and cultured. FGF22 was applied into the axonal compartment at 2DIV, and cells were fixed and stained at 8DIV. (**F**) Schematic illustration of the microfluidic chamber. Representative pictures in (**G**) and (**H**) are taken from the boxed areas. (**G–I**) Axonal treatment of FGF22 increases IGF2 in the cell body of DGCs. (**G**) Lower magnification views of Prox1 positive DGCs. (**H**) Higher magnification views from the boxed areas in (**G**). Quantification of IGF2 immunoreactivity in the cell bodies of DGCs is shown in (**I**). Error bars are s.e.m. Data are from (**B**) 292–260 cells from 3 independent experiments, (**E**) 25–45 cells from 4 to 5 independent experiments, and (**I**) 67–78 cells from 3–4 independent experiments. Significant difference from control at \*\*\*p<0.0001 by Student's t-test. Scale bars, (**A**, **C**, **D** and **H**) 10 μm, (**F**) 100 μm, (**G**) 50 μm.

*Figure 3 continued on next page*

*Figure 3 continued*

The following figure supplement is available for figure 3:

**Figure supplement 1.** A majority of DGCs in culture are calretinin-positive.

results are consistent with the notion that IGF2 is secreted from the presynaptic terminal and binds to IGF2R, which is also localized at the presynaptic terminal.

We next determined whether IGF2 expressed in DGCs promotes presynaptic development. To detect presynaptic development, we cotransfected IGF2 with synaptophysin-YFP. Overexpression of IGF2 increased the density and the size of synaptophysin-YFP puncta in DGCs compared to those in control cultures (*Figure 5A*), without apparently altering the morphology of DGCs (*Figure 5—figure supplement 1*). No effect of IGF2 overexpression was found in synaptophysin-YFP puncta in Prox1-negative non-DGCs (*Figure 5B*). These results indicate that IGF2 promotes presynaptic development specifically in DGCs.

## Neural activity is not required for FGF22 to induce IGF2 expression, but is required for IGF2 to localize to and promote development of presynaptic terminals

Synapse formation can be separated into two stages: the initial synaptic differentiation stage and the synapse maturation stage. Initial synaptic differentiation is usually regarded as an activity-independent step, while synaptic maturation, including synaptic growth, elimination, and stabilization, is influenced by neural activity. Our previous report identified that activity-dependent refinement of DGC–CA3 connections begins at ~P15 (*Yasuda et al., 2011*), which is around when we observed decreased IGF2 expression in *Fgf22*$^{-/-}$ mice (*Figure 1*). Thus, we next asked whether IGF2 expression, IGF2 localization, and/or IGF2 function for synaptogenesis require neural activity. With FGF22 treatment, IGF2 expression increased in the soma of cultured DGCs (*Figure 3A*, *Figure 6A and B*). Blockade of neural activity with tetrodotoxin (TTX) did not disturb the ability of FGF22 to increase IGF2 expression (*Figure 6A and B*). We then examined the effect of TTX on presynaptic localization of IGF2. We found that TTX treatment reduced the clustering and synaptic localization of IGF2 (*Figure 4*, *Figure 6C and D*). Finally, we assessed the requirement of neural activity in IGF2 function to induce presynaptic development. IGF2 overexpression increased synaptophysin-YFP accumulation in cultured DGCs (*Figure 5A*, *Figure 6E and F*). This increase was completely blocked by TTX treatment (*Figure 6E and F*). These results indicate that neural activity is not required for FGF22 to induce IGF2 expression, but is necessary for IGF2 to localize to and organize development of presynaptic terminals. To confirm that the activity of presynaptic neurons is critical for the localization and function of IGF2, we suppressed intrinsic neuronal excitability of DGCs by sparsely transfecting the inwardly rectifying potassium channel, Kir2.1, in culture (*Johnson-Venkatesh et al, 2015*). Similarly to the results with TTX, Kir2.1 expression in DGCs decreased the synaptic localization of IGF2 (*Figure 6G and H*) and completely blocked the synaptogenic function of IGF2 (*Figure 6I and J*). These results indicate that intrinsic neuronal excitability of DGCs is required for the presynaptic localization of IGF2 and its synaptogenic function.

## IGF2 rescues the excitatory presynaptic defects on the dendrite of *Fgf22*$^{-/-}$ CA3 pyramidal neurons

We next asked whether IGF2 acts downstream of FGF22 signaling in the regulation of excitatory presynaptic development. Our previous analysis showed that in CA3 of *Fgf22*$^{-/-}$ mice, synaptic connections are made, but synaptic vesicles fail to appropriately accumulate to the presynaptic terminals (*Terauchi et al., 2010*). In *Fgf22*$^{-/-}$ cultures, the accumulation of glutamatergic synaptic vesicles, as assessed by the number and size of VGLUT1 (vesicular glutamate transporter 1) puncta, onto dendrites of CA3 pyramidal neurons is specifically impaired (*Terauchi et al., 2010*; also see *Figure 7A*). We examined whether application of IGF2 could rescue the defects in synaptic vesicle accumulation in *Fgf22*$^{-/-}$ cultures. In WT cultures, bath application of IGF2 increased the clustering of glutamatergic synaptic vesicles on the dendrites of CA3 pyramidal neurons at 13DIV (*Figure 7A*); consistent with this result, IGF2 overexpression increased synaptophysin-YFP puncta in DGCs (*Figure 5A*,

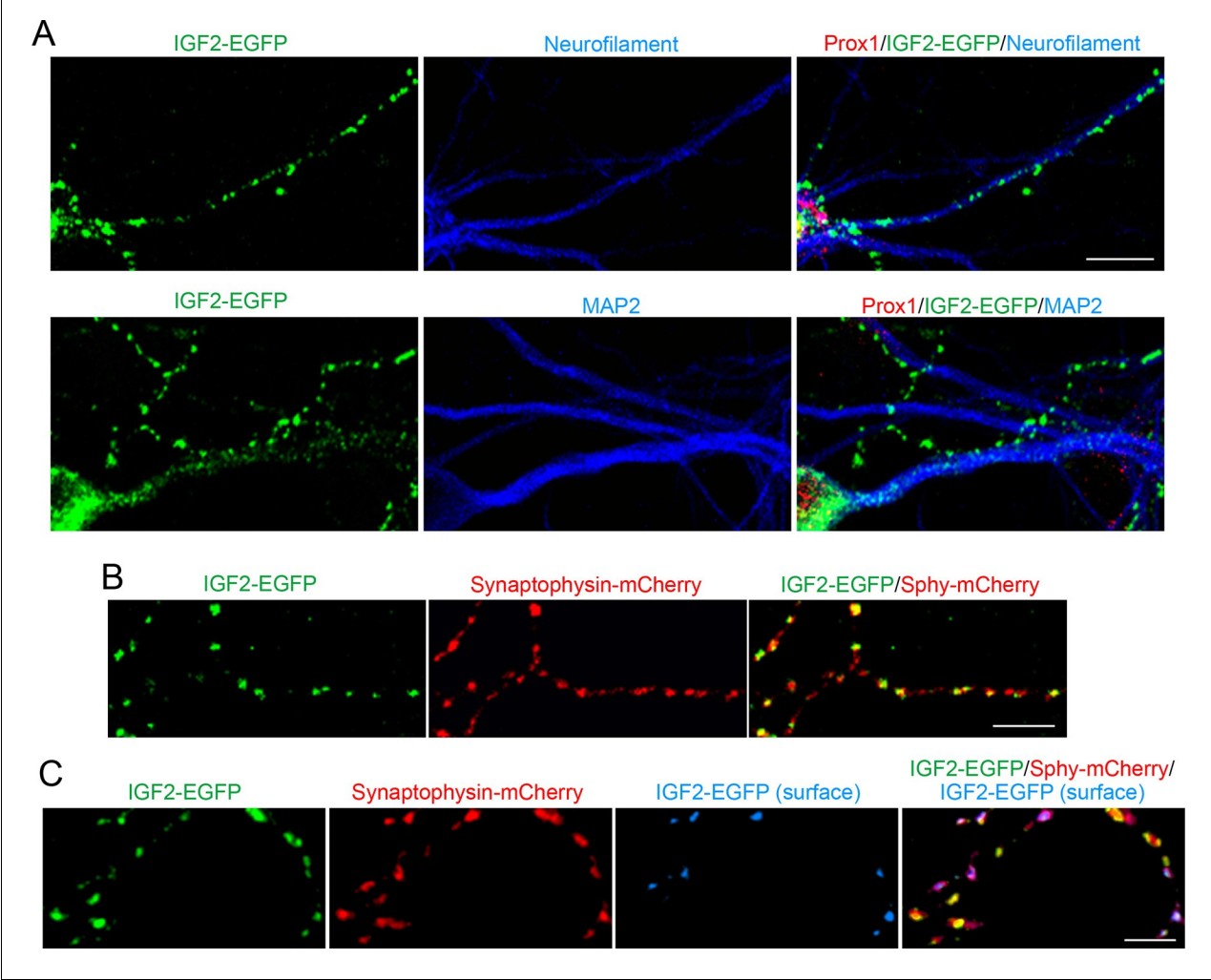

**Figure 4.** IGF2 localizes to presynaptic terminals of DGCs. Cultured hippocampal neurons were transfected with the IGF2-EGFP plasmid at 3DIV, and fixed and stained at 10DIV. (**A**) IGF2-EGFP showed a punctate pattern of localization in neurofilament positive axons, while a diffuse pattern in MAP2 positive dendrites of DGCs. Observations were from at least 10 transfected Prox1 positive DGCs from 2 independent experiments. (**B**) The IGF2-EGFP plasmid was co-transfected with synaptophysin-mCherry (Sphy-mCherry) plasmid. Most of IGF2-EGFP puncta (83.1 ± 1.0% of total IGF2-EGFP puncta; data are from 53 cells from 3 independent experiments) co-localizes with synaptophysin-mCherry, a presynaptic terminal marker. (**C**) ~40% of IGF2-EGFP is localized on the surface of presynaptic terminals. At 10DIV, cells were stained with the anti-GFP antibody without a detergent, followed by Alexa Fluor 647 (shown in blue in the images). Surface IGF2-EGFP is always colocalized with synaptophysin-mCherry. Data are from 15 cells from 3 independent experiments. Scale bars, 10 μm.

The following figure supplement is available for figure 4:

**Figure supplement 1.** IGF2R, the major receptor for IGF2, is localized at presynaptic terminals of DGCs.

*Figure 6E and F*). In *Fgf22*[-/-] cultures, IGF2 application rescued the defects in VGLUT1 accumulation: the restored number and size of VGLUT1 puncta were comparable to those seen in IGF2-treated WT cultures (*Figure 7A*). IGF2 treatment, as well as deficiency of FGF22, did not alter the clustering of VGAT (vesicular GABA transporter; a marker of GABAergic synaptic vesicles) on the dendrite of CA3 pyramidal neurons, indicating that IGF2 and FGF22 are not involved in inhibitory presynaptic development on CA3 pyramidal neurons (*Figure 7B*). In addition, IGF2 treatment and loss of FGF22 did not change the accumulation of postsynaptic markers, PSD95 (glutamatergic) and gephyrin (GABAergic), associated with CA3 pyramidal neurons (*Figure 7C and D*). Altogether, IGF2 specifically rescues excitatory presynaptic defects in *Fgf22*[-/-] neurons, suggesting that IGF2 is a mediator

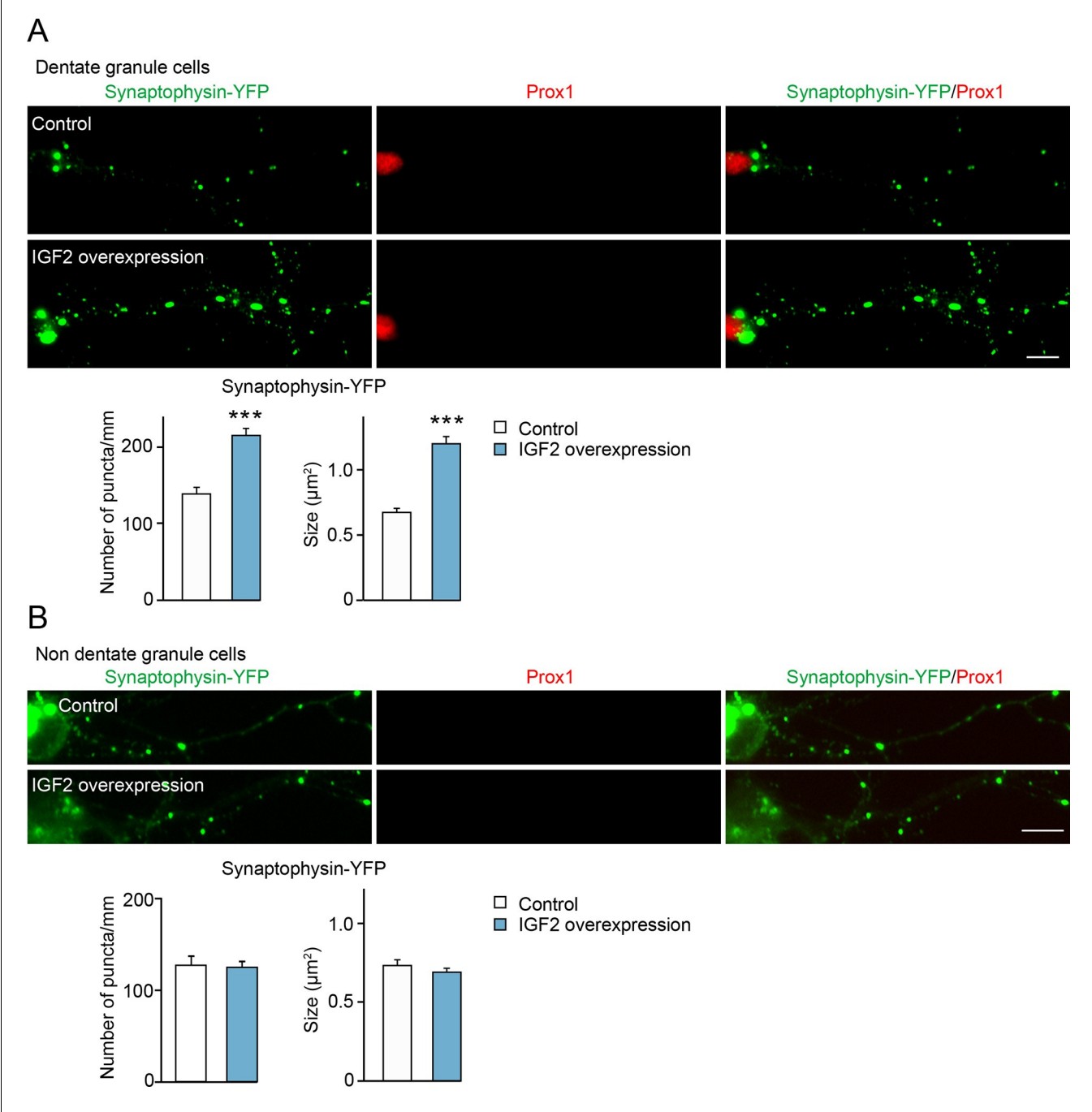

**Figure 5.** Overexpression of IGF2 in DGCs promotes their presynaptic development. Cultured hippocampal neurons were transfected with the plasmid expressing IGF2 together with the synaptophysin-YFP plasmid at 3DIV, and fixed and stained at 10DIV. (**A**) Clustering of synaptophysin-YFP is increased in IGF2-overexpressed DGCs (Prox1-positive) compared to control DGCs. The graphs show quantification of the number and size of synaptophysin-YFP puncta in control and IGF2-overexpressed DGCs. (**B**) Overexpression of IGF2 does not alter clustering of synaptophysin-YFP in Prox1-negative hippocampal neurons. The graph shows quantification of the number and size of synaptophysin-YFP puncta in Prox1 negative neurons with or without IGF2 overexpression. Error bars are s.e.m. Data are from (**A**) 36–44 cells from 5–7 independent experiments, (**B**) 11–13 cells from 2–3 independent experiments. Significant difference from control at ***p<0.0001 by Student's t-test. Scale bars, 10 μm.

The following figure supplement is available for figure 5:

**Figure supplement 1.** Overexpression of IGF2 does not appear to alter the morphology of DGCs.

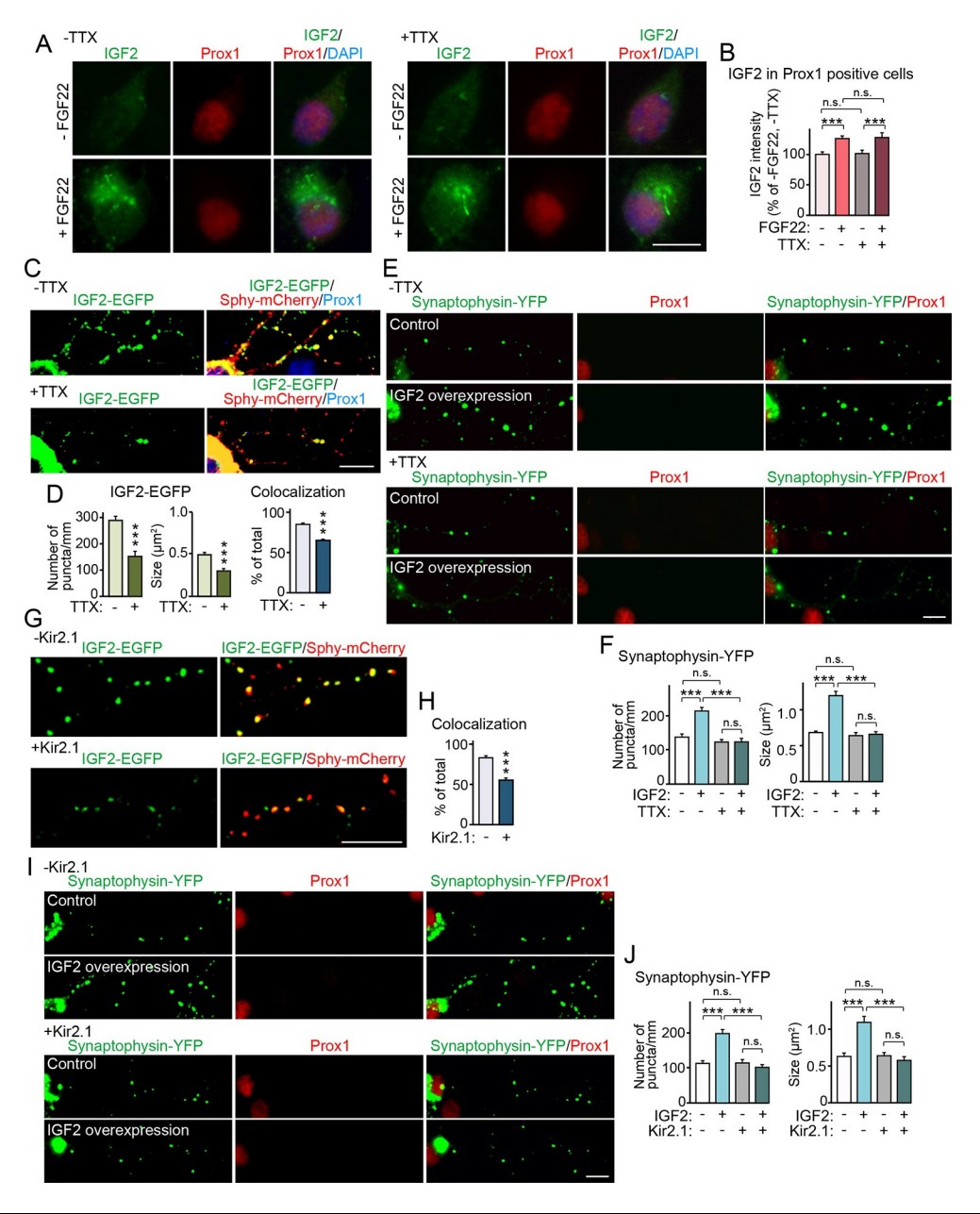

**Figure 6.** Neural activity is necessary for synaptic localization and synaptogenic effects of IGF2, but not for FGF22-dependent IGF2 expression. (**A** and **B**) Cultured hippocampal neurons were treated with FGF22 at 1DIV, and fixed and stained at 7DIV, as in *Figure 3A*. At 1DIV and 5DIV, TTX was added in the media to block neuronal activity. Bath application of FGF22 increases IGF2 expression in DGCs without (left panels) as well as with activity blockade (right panels). Quantification of IGF2 immunoreactivity in the cell bodies of DGCs is shown in (**B**). (**C–F**) TTX treatment impairs synaptic localization of IGF2 and its synaptogenic function. (**C** and **D**) Cultured hippocampal neurons were transfected with the IGF2-EGFP plasmid together with synaptophysin-mCherry (Sphy-mCherry) plasmid at 3DIV, and fixed and stained at 10DIV, as in *Figure 4B*. At 3DIV and 7DIV, TTX was added in the media to block global neuronal activity. Quantification of the number and size of IGF2-EGFP puncta, and percentage of IGF2-EGFP puncta that colocalized with synaptophysin-mCherry puncta are shown in (**D**). TTX treatment decreases IGF2-EGFP clustering and synaptic localization. (**E** and **F**) Cultured hippocampal neurons were transfected with the plasmid expressing IGF2 together with the synaptophysin-YFP plasmid at 3DIV, and fixed and stained at 10DIV, as in *Figure 5*. TTX was added in the media at 3DIV and 7DIV. IGF2 overexpression increases clustering of synaptophysin-YFP in Prox1-positive DGCs without TTX (upper panels), but not with TTX (lower panels). (**F**) Quantification of the number and size of synaptophysin-YFP in DGCs with or without IGF2 overexpression in the

*Figure 6 continued on next page*

*Figure 6 continued*

presence or absence of TTX. (**G**–**J**) Suppression of intrinsic neuronal excitability impairs synaptic localization of IGF2 and its synaptogenic function. (**G** and **H**) Cultured hippocampal neurons were transfected with the plasmids expressing Kir2.1, IGF2-EGFP, and synaptophysin-mCherry (Sphy-mCherry) at 3DIV. Cells were fixed and stained at 10DIV. Percentage of IGF2-EGFP puncta that colocalized with synaptophysin-mCherry puncta in Prox1-positive DGCs is shown in (**H**). Kir2.1 expression decreases synaptic localization of IGF2-EGFP in DGCs. (**I** and **J**) Cultured hippocampal neurons were transfected with the plasmids expressing Kir2.1, IGF2, and synaptophysin-YFP at 3DIV. Cells were fixed and stained at 10DIV. (**J**) Quantification of the number and size of synaptophysin-YFP in DGCs with or without IGF2 overexpression in the presence or absence of Kir2.1. IGF2 overexpression increases clustering of synaptophysin-YFP in Prox1-positive DGCs, but the increase is suppressed by Kir2.1 expression. Error bars are s. e.m. Data are from (**B**) 35–292 cells from 3 independent experiments, (**D**) 20–32 cells from 3–4 independent experiments, (**F**) 22–24 cells from 3 independent experiments, (**H**) 7 cells from 3 independent experiments, (**J**) 20 cells from 4 independent experiments. Significant difference from control at ***$p < 0.0001$ by two-way ANOVA followed by Tukey's multiple comparison test (**B**, **F**, and **J**) and by Student's t-test (**D** and **H**). n.s.: no statistical significant difference. Scale bars, 10 μm.

of FGF22 signaling to promote excitatory presynaptic development on the dendrites of CA3 pyramidal neurons.

## IGF2 is not required for FGF22-induced initial presynaptic differentiation, but is necessary for subsequent presynaptic stabilization

We then asked whether IGF2 is necessary for FGF22-dependent excitatory presynaptic development. Defects in excitatory synapse formation are observed in the CA3 region of the hippocampus in *Fgf22$^{-/-}$* mice as early as P8, an early stage of synapse formation (*Terauchi et al., 2010*). Meanwhile, decreased IGF2 expression was observed in *Fgf22$^{-/-}$* mice at P14, but not at P7 (*Figure 1*), and the effects of IGF2 on presynaptic development is activity-dependent (*Figure 6C–F*). These results raise the possibility that IGF2 contributes to FGF22-dependent presynaptic development in a stage-specific manner: IGF2 is not required during the initial stage but is required during the following stage of synapse formation, i.e., activity-dependent synapse stabilization. To test this possibility, we employed an shRNA knockdown approach (*Figure 8—figure supplement 1*) to silence the expression of IGF2 in DGCs from 1DIV by using two independent shRNA plasmids. Knockdown of IGF2 did not apparently alter the morphology of DGCs (*Figure 8—figure supplement 2*). We assessed synaptic vesicle (synaptophysin-YFP) accumulation in response to FGF22 in the axons of IGF2-knockdown DGCs (from 1DIV) during the initial presynaptic differentiation (6DIV) and subsequent presynaptic stabilization (12DIV) stages. At 6DIV, we did not observe any differences between IGF2-knockdown and control DGCs: the basal levels of synaptophysin-YFP accumulation were not different, and IGF2-knockdown DGCs still responded to FGF22 treatment to increase the accumulation of synaptophysin-YFP (*Figure 8A and B*). On the other hand, at 12DIV, the number and size of synaptophysin-YFP puncta were significantly decreased in IGF2-knockdown DGCs compared to those in controls (*Figure 8C and D*). In addition, IGF2-knockdown DGCs no longer show FGF22-dependent increases in the accumulation of synaptophysin-YFP (*Figure 8C and D*). These results suggest that IGF2 is not necessary for the initial presynaptic differentiation induced by FGF22, but is required for a later stage of presynaptic development. To confirm the role of IGF2 in the late stage of presynaptic development, we knocked down IGF2 from 6DIV and assessed presynaptic development at 12DIV. IGF2 knockdown from 6DIV decreased the number and size of presynaptic terminals (as assessed by synaptophysin-YFP accumulation) and blocked the synaptogenic effects of FGF22 at 12DIV (*Figure 8E and F*), suggesting that IGF2 is indeed critical for the later stage of presynaptic stabilization.

## *Igf2$^{-/-}$* mice show defects in DGC presynaptic development in the stabilization stage and not in the initial stage of synapse formation

Finally, to investigate whether and at which developmental stages IGF2 is necessary for excitatory presynaptic development in vivo, we examined clustering of excitatory synaptic vesicles in CA3 of *Igf2$^{-/-}$* mice from P8 to P29. In the hippocampus, synapse development from P0 to ~P14 is not

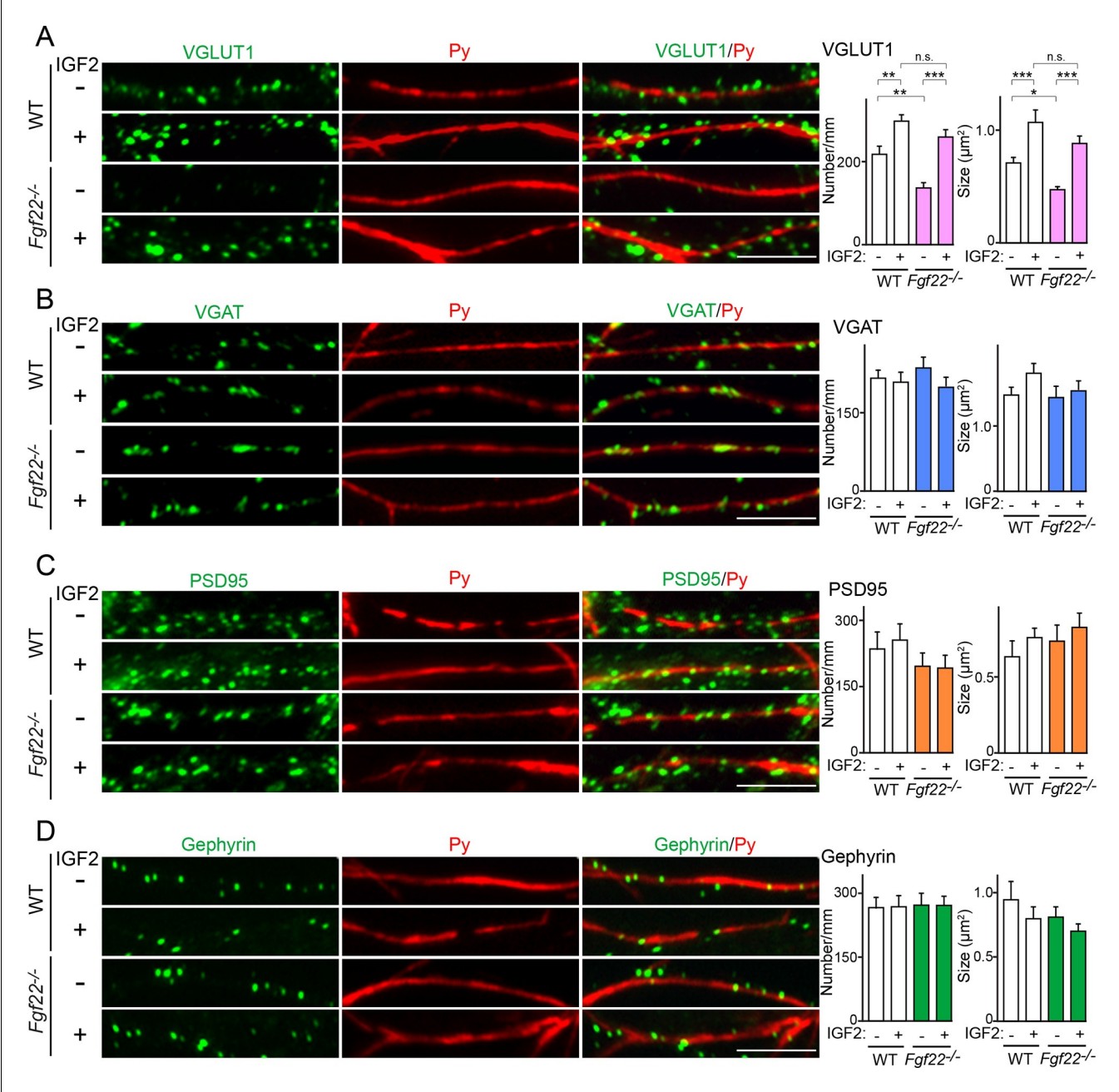

**Figure 7.** IGF2 treatment rescues the impairment of glutamatergic presynaptic development in *Fgf22*^-/- cultures. Hippocampal neurons from WT and *Fgf22*^-/- mice were cultured with or without IGF2 treatment. IGF2 was applied into culture media at 1DIV, and cells were fixed and stained at 13DIV. (A) VGLUT1 clustering on the dendrites of CA3 pyramidal neurons (immunolabeled with Py antibody). IGF2 treatment increases VGLUT1 clustering in WT culture and rescues defects in VGLUT1 clustering in *Fgf22*^-/- cultures. (B–D) Application of IGF2 does not change VGAT clustering (B), PSD95 clustering (C), and gephyrin clustering (D) on the dendrites of CA3 pyramidal neurons in WT and *Fgf22*^-/- cultures. The density (number/mm) and size of VGLUT1, VGAT, PSD95 and gephyrin puncta on CA3 pyramidal neurons were analyzed and shown in the graphs. Error bars are s.e.m. Data are from 12–39 cells from 3–7 independent experiments. Significant difference from control at *$p<0.05$, **$p<0.01$ and ***$p<0.001$ by two-way ANOVA followed by Tukey's multiple comparison test. n.s.: no statistical significant difference. Scale bars, 10 μm.

apparently influenced by neural activity ("initial synapse differentiation"), but that from ~P14 to ~P28 is regulated by activity, where activity-dependent synapse maturation (e.g., *Toth et al., 2013*) and activity-dependent synapse elimination (*Yasuda et al., 2011*) take place. We found that at P8, clustering of VGLUT1 in CA3 of *Igf2*^-/- mice was similar to that of WT mice (*Figure 9A–D*). At P14,

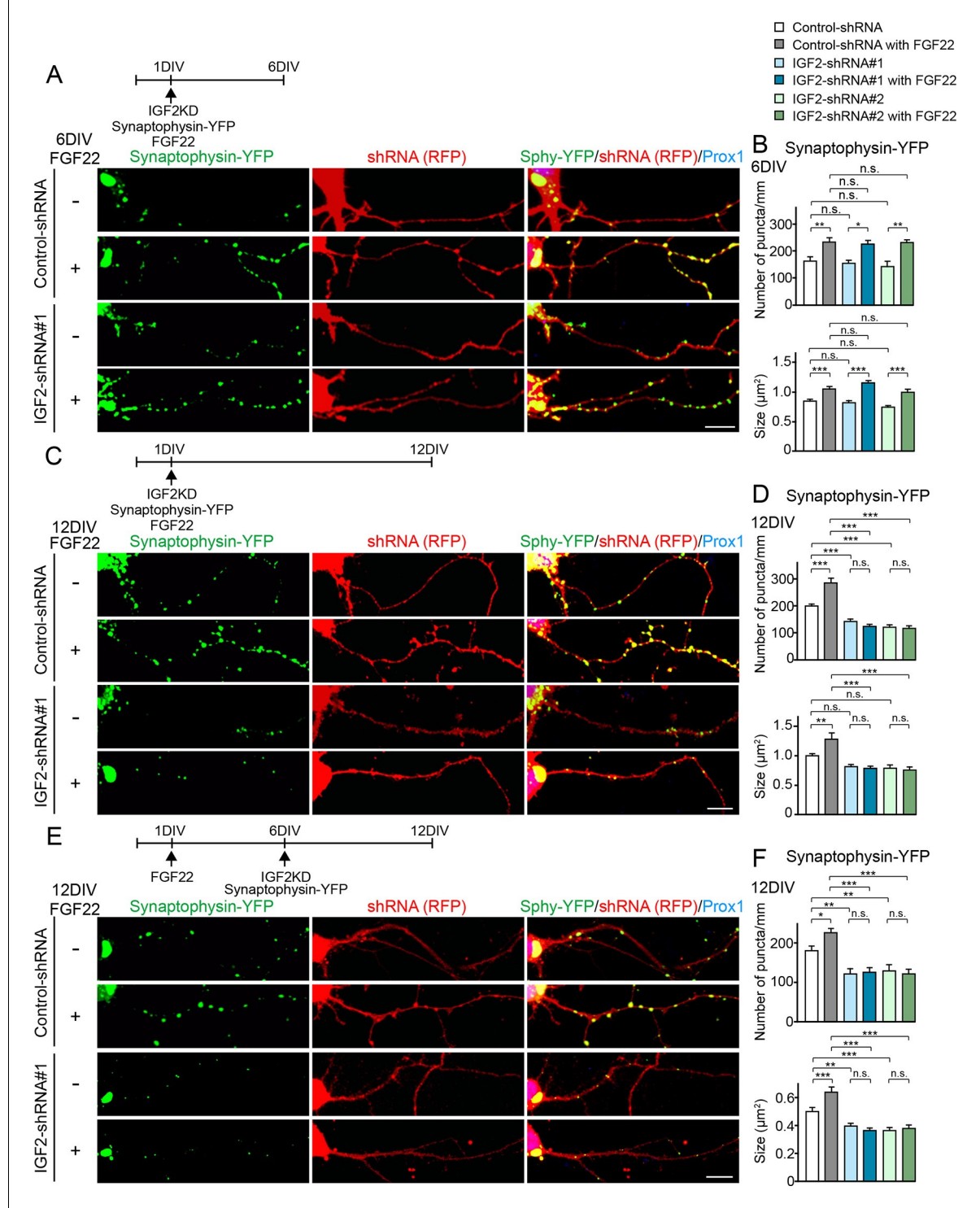

**Figure 8.** IGF2 is dispensable for FGF22-dependent initial presynaptic differentiation, but is required for subsequent presynaptic stabilization. (**A-D**) Cultured hippocampal neurons were transfected with the plasmid expressing synaptophysin-YFP together with the plasmid expressing either control-shRNA, IGF2-shRNA#1, or IGF2-shRNA#2 at 1DIV. FGF22 was applied into culture media after the transfection at 1DIV. Cells were fixed, stained, and clustering of synaptophysin-YFP in DGCs was analyzed at 6DIV or 12DIV. (**A** and **B**) Clustering of synaptophysin-YFP in DGCs at 6DIV. Quantification of the number and size of synaptophysin-YFP clusters in DGCs are shown in the graph (**B**). FGF22 treatment increases the number and size of synaptophysin-YFP puncta in both control and IGF2-knockdown DGCs. (**C** and **D**) Clustering of synaptophysin-YFP in DGCs at 12DIV. Quantification of the number and size of synaptophysin-YFP clusters are shown in the graph (**D**). Without IGF2, the effects of FGF22 on synaptophysin-YFP clustering disappear. (**E** and **F**) Cultured hippocampal neurons were treated with FGF22 at 1DIV. At 6DIV, neurons were transfected with the plasmid expressing

*Figure 8 continued on next page*

Figure 8 continued

synaptophysin-YFP together with the plasmid expressing either control-shRNA, IGF2-shRNA#1, or IGF2-shRNA#2. Cells were fixed and stained at 12DIV. Quantification of the number and size of synaptophysin-YFP clusters are shown in the graph (F). IGF2-knockdown at a late stage of synapse development (from 6DIV) decreases synaptophysin-YFP clustering and blocks the synaptogenic effects of FGF22 in DGCs. Error bars are s.e.m. Data are from 12–62 cells from 5–8 independent experiments. Significant difference from control at *$p<0.05$, **$p<0.01$ and ***$p<0.001$ by two-way ANOVA followed by Tukey's multiple comparison test. n.s.: no statistical significant difference. Scale bars, 10 μm.

The following figure supplements are available for figure 8:

**Figure supplement 1.** Efficiency of shRNA-mediated IGF2 knockdown.

**Figure supplement 2.** Effects of shRNA-mediated IGF2 knockdown on DGC morphology.

*Igf2$^{-/-}$* mice still had a similar number of VGLUT1 puncta, but their size was smaller in the CA3 stratum lucidum (SL) region, where the axons of DGCs form excitatory synapses with CA3 pyramidal neurons. At P21 and P29, *Igf2$^{-/-}$* mice showed a significant decrease in the number and size of VGLUT1 puncta in the SL layer (*Figure 9A–C*). The targeting of DGC axons to CA3 appeared normal in *Igf2$^{-/-}$* mice (*Figure 9—figure supplement 1*). No defects were observed in the CA3 stratum radiatum (SR) region, where CA3 to CA3 synapses are located, throughout the time periods we examined (*Figure 9A and D*). These results suggest that IGF2 is critical for a later stage (after P14) of DGC–CA3 synapse development, but not for CA3–CA3 synapse development.

To further understand the role of IGF2 in synaptic development, we analyzed ultrastructure of excitatory (asymmetric) synapses in the SL and SR layers of CA3 in *Igf2$^{-/-}$* mice at P28–P29. Both in the SL and SR layers, the number and size of postsynaptic densities were similar between WT and *Igf2$^{-/-}$* mice (*Figure 10*). In the presynaptic terminals in the SL layer of *Igf2$^{-/-}$* mice, there were fewer synaptic vesicles, less clustering of synaptic vesicles, fewer docked vesicles, and smaller synaptic vesicles relative to WT mice (*Figure 10A and B*). In contrast, no structural defects were found in the SR layer of *Igf2$^{-/-}$* mice (*Figure 10C and D*). Together with the immunostaining results (*Figure 9*), our results suggest that in *Igf2$^{-/-}$* mice, synaptic vesicles were not stabilized/maintained in the presynaptic terminals specifically at the DGC–CA3 synapses.

To address the physiological consequences of IGF2 deficiency in synaptic function, we recorded synaptic current from the CA3 region of the hippocampus from adult WT and *Igf2$^{-/-}$* mice. The frequency but not the amplitude of miniature excitatory postsynaptic currents (mEPSCs) was significantly decreased in CA3 pyramidal neurons of adult *Igf2$^{-/-}$* mice (*Figure 11A and B*), suggesting that loss of IGF2 has a prolonged impact on excitatory presynaptic function, without significantly affecting postsynaptic function. To examine whether synaptic defects are specific to DGC–CA3 connections, we recorded evoked field excitatory postsynaptic potentials (fEPSPs) at DGC–CA3 or CA3–CA3 synapses (*Figure 11C*). fEPSP responses were verified as DGC–CA3 or CA3–CA3 based on the sensitivity to DCG-IV, the group 2 mGluR agonist that selectively blocks DGC–CA3 responses (red traces in *Figure 11C*; *Nicoll and Schmitz, 2005*). The fEPSP slope was significantly smaller in *Igf2$^{-/-}$* than WT mice for DGC–CA3 responses (*Figure 11C*), but not for CA3–CA3 responses. Paired-pulse facilitation at DGC–CA3, but not CA3–CA3, synapses was decreased in *Igf2$^{-/-}$* mice relative to WT mice (*Figure 11D*). Taken together, these results suggest that excitatory synaptic transmission at DGC–CA3 synapses is specifically impaired in *Igf2$^{-/-}$* mice due to the loss of synaptic vesicles from the presynaptic terminals.

## Discussion

In this study, we identified *Igf2* as a target gene induced by the presynaptic organizer FGF22 and showed that in the mammalian hippocampus: i) IGF2 localizes to presynaptic terminals of DGCs and promotes their maturation in an activity-dependent manner; ii) IGF2 can rescue excitatory presynaptic defects in *Fgf22$^{-/-}$* neurons; iii) IGF2 is required for FGF22-dependent presynaptic stabilization, but not for initial presynaptic differentiation; and iv) *Igf2$^{-/-}$* mice show presynaptic defects at DGC–CA3 synapses during synaptic stabilization. Together, we propose a novel, activity-dependent feedback signaling pathway for presynaptic stabilization in the mammalian brain: target-derived FGF22

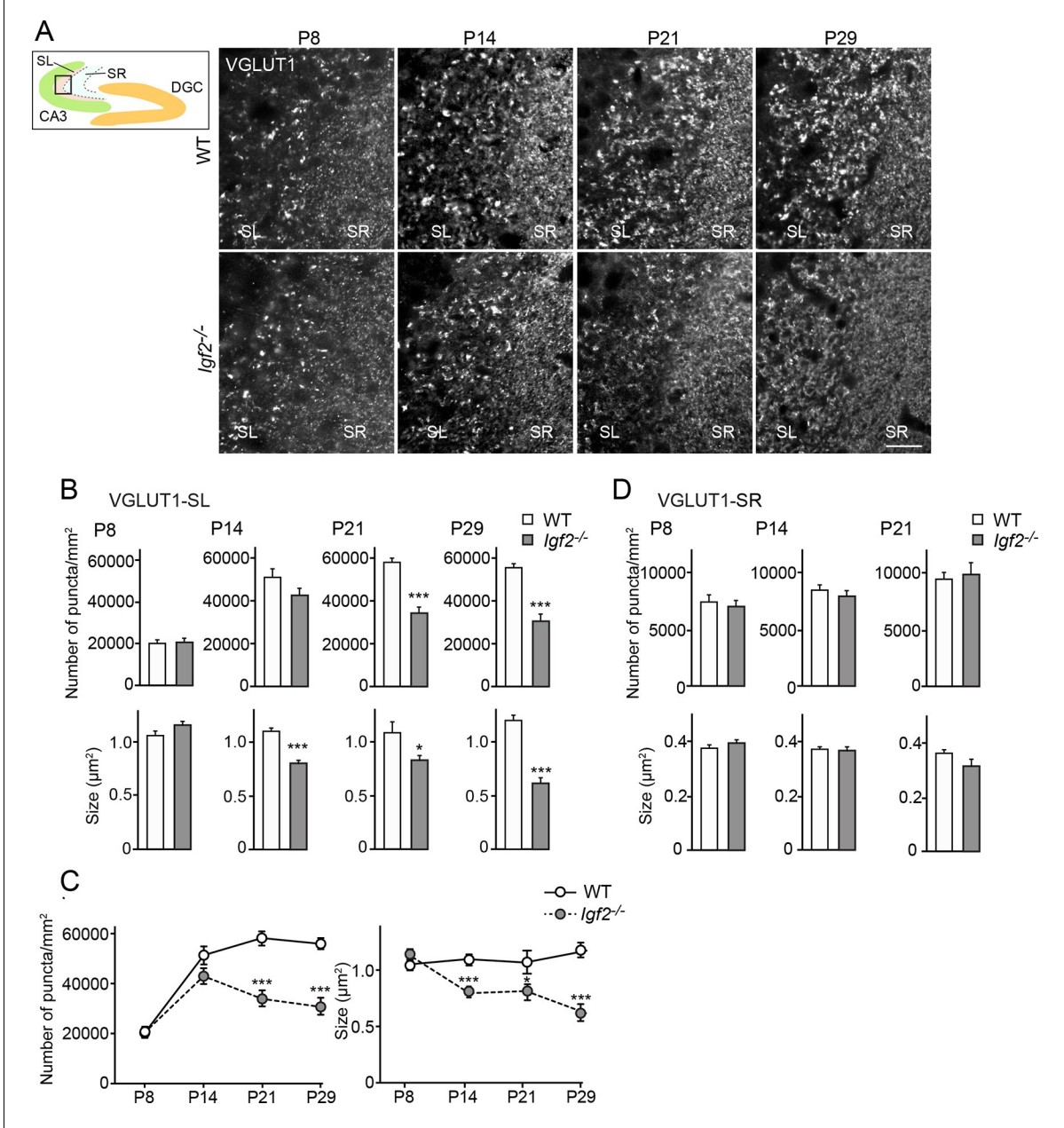

**Figure 9.** Glutamatergic presynaptic stabilization is impaired at the DGC–CA3 synapses in *Igf2*$^{-/-}$ mice. Hippocampal sections from WT and *Igf2*$^{-/-}$ mice at P8, P14, P21, and P29 were immunostained for VGLUT1. The illustration shows the pictured area (boxed). (**A**) Representative pictures from CA3 regions. SL: stratum lucidum (DGC–CA3 synapses); SR: stratum radiatum (CA3–CA3 synapses). Density and size of VGLUT1 puncta are quantified and shown in (**B**: SL) and (**D**: SR). (**C**) Time course of VGLUT1 clustering in the SL layer. In the SL layer, *Igf2*$^{-/-}$ mice show no presynaptic defects at P8, but start to show mild defects at P14, and exhibit significant defects at P21 and P29. No significant defects were observed in the SR layer. Error bars are s. e.m. Data are from (**B** and **C**) 23–42 fields from 3–5 independent experiments, (**D**) 13–34 fields from 3–5 independent experiments. Significant difference from control at *p<0.05 and ***p<0.0001 by Student's t-test. Scale bar, 20 μm.

The following figure supplement is available for figure 9:

**Figure supplement 1.** The targeting of DGC axons to CA3 appears normal in *Igf2*$^{-/-}$ mice.

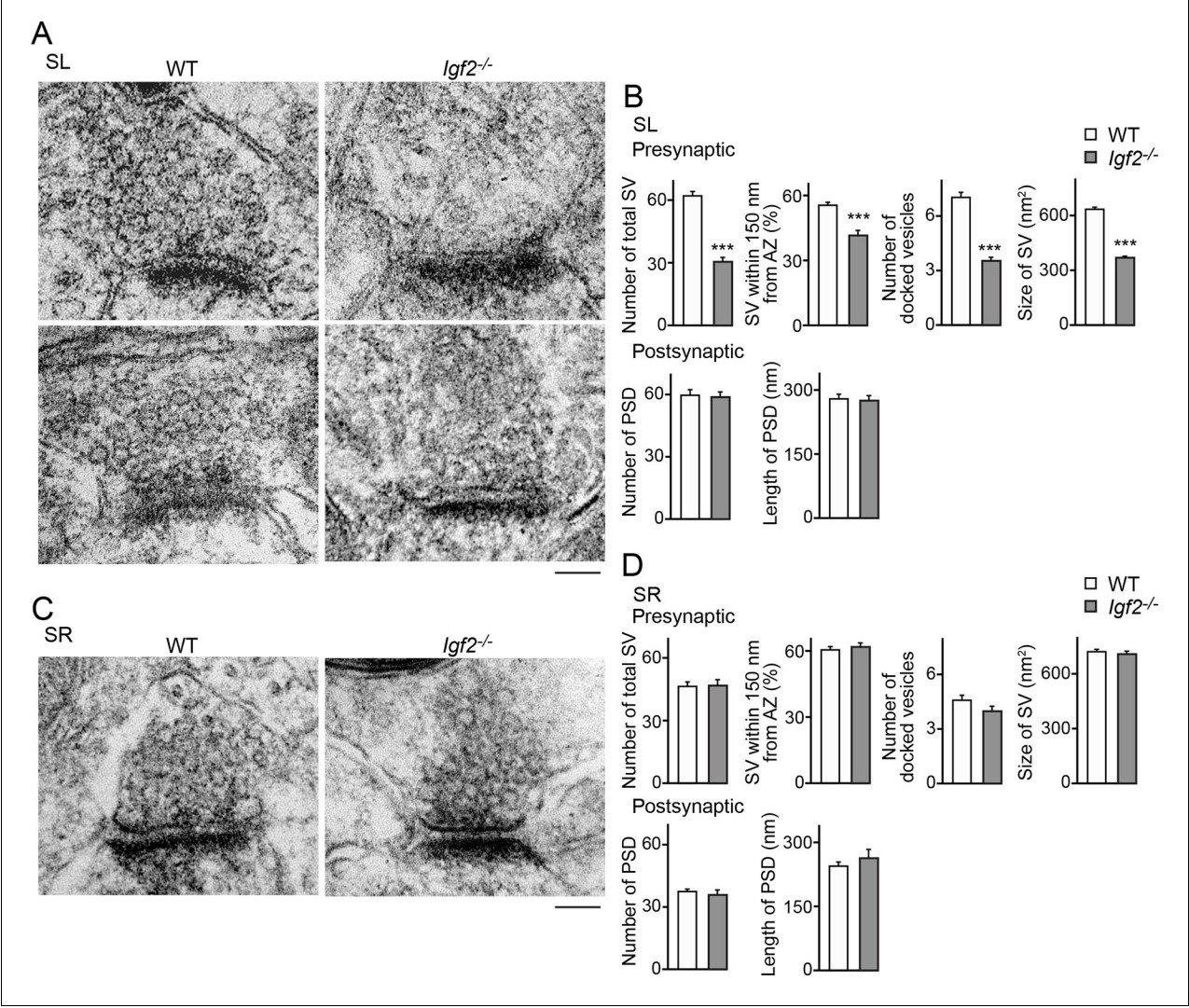

**Figure 10.** Electron microscopic analysis shows defects in excitatory presynaptic terminals selectively in the SL layer of *Igf2*[-/-] mice. Electron microscopic analysis of asymmetric (excitatory) synapses in WT and *Igf2*[-/-] mice (P28–29). (**A** and **B**) Asymmetric synapses in the CA3 SL layer. (**A**) Two representative images of asymmetric synapses in WT and *Igf2*[-/-] mice. (**B**) Quantification of synaptic vesicles (SVs) and postsynaptic densities (PSDs). Number of SVs within 400 nm from the active zone (total SV), % SVs within 150 nm from active zone, number of docked vesicles per synapse, size of SVs, number of PSDs in 100 nm[2], and length of PSDs are shown. (**C** and **D**) Asymmetric synapses in the CA3 SR layer. (**C**) Representative images of asymmetric synapses in WT and *Igf2*[-/-] mice. (**D**) Quantification of SVs and PSDs. *Igf2*[-/-] mice show a loss of SVs in the SL layer, but not in the SR layer, without apparent changes in PSDs. Error bars indicate s.e.m. Data are from 14–50 synapses from 12–13 fields from 2 mice per strain. Significant differences from WT mice at ***p<0.0001 by Student's t-test. Scale bars, 100 nm.

induces the presynaptic expression of IGF2, and IGF2 in turn, localizes to the presynaptic terminals in an activity-dependent manner for their stabilization (*Figure 12*).

## A novel feedback pathway through transcriptional regulation for presynaptic stabilization in the mammalian brain

In *Drosophila* motor neurons, target-derived Gbb signaling induces the expression of *Trio* in the presynaptic neurons; Trio is then transported back to the nerve terminal and stabilizes cytoskeletal structures of nerve terminals (*Ball et al., 2010*). In contrast, in the mammalian brain, such feedback pathways for presynaptic stabilization had not been identified. We focused on FGF signaling, because gene expression is one of the most significant outcomes of FGF signaling (*Dorey and Amaya, 2010*; *Partanen, 2007*). Thus, FGF-dependent gene expression is poised to have important

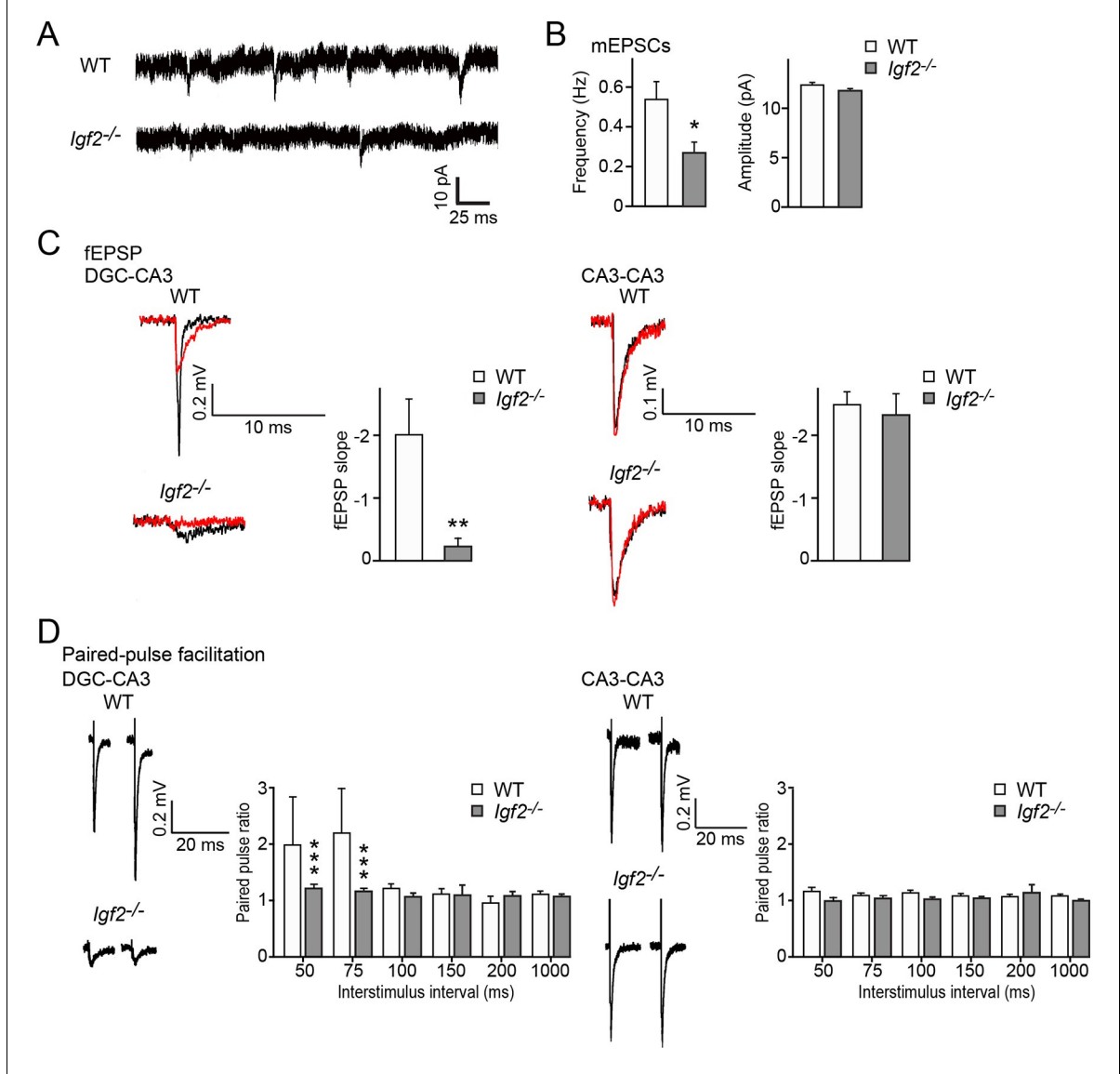

**Figure 11.** Excitatory synaptic transmission at DGC–CA3 synapses, but not CA3–CA3 synapses, is specifically impaired in *Igf2*[-/-] mice. (**A** and **B**) Whole-cell recordings of mEPSCs from CA3 pyramidal neurons of adult WT and *Igf2*[-/-] hippocampal slices. (**A**) Representative traces of mEPSCs. (**B**) Quantification of the frequency and amplitude of mEPSCs. The frequency, but not amplitude, of mEPSCs is specifically decreased in CA3 of *Igf2*[-/-] mice. 31–32 cells from 5 mice per genotype. *p<0.05 by Student's t-test. (**C**) fEPSP responses evoked in the CA3 region of the hippocampus. Responses were characterized as either DGC–CA3 or CA3–CA3 based upon sensitivity to DCG-IV treatment. Black traces: original responses, red traces: after 10 min of DCG-IV treatment. Graphs show the quantification of the maximum fEPSP slope. At DGC–CA3 synapses, the maximum elicited response is significantly smaller in *Igf2*[-/-] mice than in WT mice (n = 8 and 8; **p<0.01 by Student's t-test). At CA3–CA3 synapses, the maximum elicited response is not different between WT and *Igf2*[-/-] mice (n = 14 and 24; p = 0.71 by Student's t-test). (**D**) Paired pulse facilitation (PPF). PPF at the DGC–CA3 synapses, and not CA3–CA3 synapses, is significantly decreased in *Igf2*[-/-] mice (***p<0.0001 by Two-way ANOVA followed by a Tukey test). Example traces demonstrate responses with a 50 ms inter-stimulus interval.

effects on neuronal network development. Here, we have shown that the FGF22–IGF2 signaling serves as a feedback pathway important for the stabilization of DGC presynaptic terminals in the mammalian hippocampus.

An important next question is how IGF2 stabilizes presynaptic terminals. IGF2, which often acts as an autocrine factor (*Pollak et al., 2004*), mediates its functions mainly via IGF2 receptor (IGF2R) as well as via IGF1 receptor (IGF1R) (*Fernandez and Torres-Aleman, 2012*). In neurons, both receptors are used to mediate IGF2 signal and participate in various physiological functions (*Agis-*

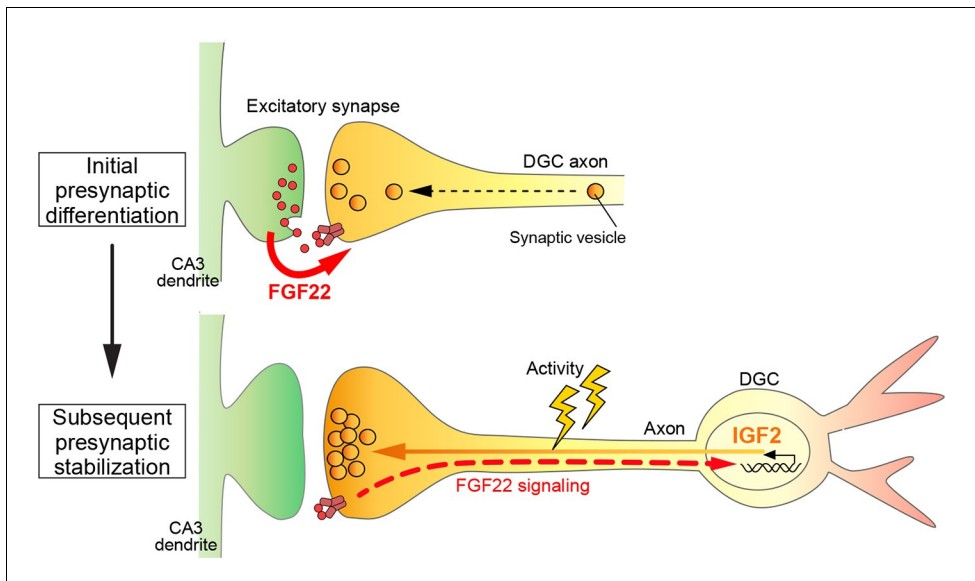

**Figure 12.** A Model showing a feedback signal for activity-dependent presynaptic stabilization. For initial presynaptic differentiation, target-derived FGF22 locally promotes clustering of synaptic vesicles. For subsequent presynaptic stabilization, FGF22 signaling induces the expression of IGF2. IGF2, in turn, localizes to presynaptic terminals in an activity-dependent manner and stabilizes them.

*Balboa et al., 2011*; *Chen et al., 2011*; *Schmeisser et al., 2012*). We showed that IGF2 is secreted and tethered on the surface of presynaptic terminals of DGCs (*Figure 4C*). IGF2R (*Figure 4—figure supplement1*) and IGF1R (*Gazit et al., 2016*) are also localized at presynaptic terminals. These results suggest that IGF2 is secreted from the presynaptic terminal and binds to IGF2R/IGF1R, which is also localized at the presynaptic terminal, so that IGF2 acts locally in an autocrine manner for its effects. Yet, we cannot exclude the possibility that IGF2 may also act as a global presynaptic organizer if IGF2 is released from the presynaptic terminal. IGF1R is a receptor tyrosine kinase, which may regulate local translation, and IGF2R is known to signal through G proteins, which may ultimately affect calcium homeostasis. Calcium homeostasis and local translation of synaptic proteins, as well as regulation of cytoskeletal structures like Trio may all contribute to IGF2-dependent presynaptic stabilization.

## Activity-dependent regulation of presynaptic stabilization by IGF2

Neural activity is involved in synaptic stabilization. It has been proposed that active synapses are stabilized and inactive ones are destabilized (*Lichtman and Colman, 2000*; *Ruthazer and Cline, 2004*). Various forms of activity are involved: in most cases, synaptic transmission and synaptic competition are considered critical for synapse refinement (*Lichtman and Colman, 2000*; *Waites et al., 2005*). In addition, intrinsic activity also plays critical roles in synapse development (*Johnson-Venkatesh et al., 2015*). However, the molecular mechanisms by which intrinsic activity contributes to synapse stabilization during development are largely unknown. We found that IGF2's function in synaptic stabilization requires intrinsic neuronal activity (*Figure 6*). IGF2 is not transported to presynaptic terminals when intrinsic neuronal excitability of DGCs is suppressed. Thus, our results reveal that IGF2 is a mediator of intrinsic activity-dependent presynaptic stabilization. Since IGF2's effects are for the stabilization of synaptic vesicles but not for the initial recruitment of synaptic vesicles (which is carried out by FGF22), it is reasonable that TTX/Kir2.1 did not affect synaptic vesicle transportation (*Figure 6*).

How does intrinsic activity regulate IGF2 localization? Neural activity is known to control motor function of KIF proteins and regulate intracellular transport of synaptic components, such as mitochondria and AMPA receptors (*Hoerndli et al., 2015*; *Saxton and Hollenbeck, 2012*). Neural activity also modulates phosphorylation of the C terminal domain of KIF3A, which affects loading of N-

cadherin containing cargos and contributes to the maintenance of homeostatic synaptic plasticity (*Ichinose et al., 2015*). Thus, neural activity may regulate the IGF2 transport complex through phosphorylation of motor proteins. It will be interesting to investigate the mechanisms of IGF2 transport to presynaptic terminals and how neural activity regulates the transport.

In addition, neural activity may regulate the secretion of IGF2. Neural activity controls exocytotic secretion of cytoplasmic vesicles in neurons: in olfactory bulb neurons, K$^+$-induced depolarization activates the exocytotic Ca$^{2+}$-sensor, synaptotagmin-10, to induce secretion of IGF1 (*Cao et al., 2011*). Thus, in addition to transportation, secretion of IGF2 at presynaptic sites might also be controlled by neural activity. If this is the case, one may speculate that activity blockade would increase the intracellular IGF2 levels. However, we did not observe an increase in the intracellular IGF2 levels, probably because neural activity is also critical for IGF2 transportation (*Figure 6*).

It is also possible that various activity-dependent genes might influence IGF2/IGF2R localization, which is an interesting future study. Our Kir2.1 experiments, in which Kir2.1 was sparsely transfected so that we can ignore effects from postsynaptic neurons, suggest that changes in postsynaptic activity do not play critical roles in IGF2 localization and function.

## Cell-wide development of presynaptic neurons by IGF2

In addition to presynaptic stabilization, signals propagated from target-derived molecules may influence further cell-wide development of the presynaptic neurons. In the mammalian brain, signaling from target-derived neurotrophins has been well characterized. For example, nerve growth factor (NGF) signaling regulates gene expression, including *TrkA*, *p75*, *Bdnf*, and *Ntf5*, and controls cell survival and death of own and neighboring neurons (*Ascano et al., 2012*; *Deppmann et al., 2008*; *Singh et al., 2008*) as well as dendritic development (*Sharma et al., 2010*).

Since IGF2 has been shown to exhibit broad functions in the brain, IGF2 may not only contribute to presynaptic stabilization, but also various aspects of cell-wide development, including neurogenesis, neurite growth, and spine maturation. For neurogenesis, IGF2 is implicated in maintenance and expansion of neural stem cells (*Ziegler et al., 2012*; *2014*) and proliferation of neuronal progenitor cells (*Burns and Hassan, 2001*; *Lehtinen et al., 2011*) in the developing brain. IGF2 also contributes to adult neurogenesis in the subgranular zone of the hippocampus (*Bracko et al., 2012*; *Ouchi et al., 2013*). For neuronal development, IGF2 leads to nerve sprouting (*Caroni and Grandes, 1990*), neurite outgrowth (*Jeong et al., 2013*), as well as spine maturation in cultured hippocampal neurons (*Schmeisser et al., 2012*). Thus, IGF2 may serve as a general regulator of the development of presynaptic neurons (DGCs) downstream of FGF22.

Since IGF2 is still expressed in calbindin-positive DGCs in *Fgf22$^{-/-}$* mice (*Figure 1*), IGF2 can be expressed in an FGF22-independent manner as well. Indeed, IκB, which is not utilized by FGF22 signaling, has been shown to induce expression of IGF2 for spine maturation in mature hippocampal neurons (*Schmeisser et al., 2012*). The expression of the mouse *Igf2* gene is regulated by three alternative promoters (*Sasaki et al., 1992*). Thus, different signals seem to be used to regulate distinct phases of neuronal development.

## Temporal and spatial specificity of IGF2 effects

The presynaptic effects of IGF2 are stage and cell-type specific. FGF22-dependent IGF2 expression was observed in young, calretinin-positive DGCs, but not in mature, calbindin-positive DGCs (*Figures 1–3*). Specific responsiveness of calretinin-positive DGCs to FGF22 appears to be linked to the developmental stage. Calretinin-positive DGCs elongate axons to CA3 and contact CA3 pyramidal neurons to form synapses (*Aguilar-Arredondo et al., 2015*; *Li et al., 2009*; *Ming and Song, 2005*; *Yasuda et al., 2011*). Thus, around that stage, DGCs may become more responsive to FGF22. Our results are consistent with the idea that IGF2 is induced by CA3-derived FGF22 when DGC axons contact with their target dendrites. The data from CA3-selective *Fgf22* knockout mice (*Figure 2*) and the local FGF22 application experiments, in which IGF2 was induced by axonal application of FGF22 (*Figure 3F–I*), further support this idea. Our results also suggest that IGF2 is not critical for initial synaptic differentiation, because no defects were found in the absence of IGF2 at 6DIV in vitro (*Figure 8A,B*) or P8 in vivo (*Figure 9*). However, it is important for the later stages of synapse development: after 6DIV in vitro (*Figure 8E,F*) and P14 in vivo (*Figure 9*), when neural activity influences synapse formation (*Toth et al., 2013*; *Yasuda et al., 2011*). As IGF2 effects are activity

dependent (*Figure 6*), we propose that IGF2, induced by target-derived FGF22, is mainly important for the stages of activity-dependent synapse stabilization.

Interestingly, induction of IGF2 by FGF22 is specific to DGCs. CA3 pyramidal neurons, which release FGF22, receive excitatory synaptic inputs from collateral/associational CA3 pyramidal neurons and stellate cells in the entorhinal cortex, in addition to inputs from DGCs (*Urban et al., 2001*). However, CA3 neurons do not respond to FGF22 to induce IGF2 (*Figure 3*). Since *Fgf22*$^{-/-}$ mice show defects in excitatory synapse formation both in the SL (where DGC to CA3 synapses are located) as well as SR (where CA3 to CA3 synapses are located) (*Terauchi et al., 2010*), the lack of IGF2 induction in CA3 neurons indicates that IGF2 is a unique target of FGF22 specifically in differentiating DGCs. It is possible that CA3 neurons express genes other than IGF2 in an FGF22-dependent manner for presynaptic stabilization. If so, different types of presynaptic neurons might process FGF22 signaling through their own transcription regulation to promote neuron type-specific presynaptic stabilization.

It is worth noting that many synapse organizing molecules show input specificity. FGF22 affects synapse formation in the SR and SL layers of CA3, but not the SLM (stratum lacunosum moleculare) layer (*Terauchi et al., 2010*). Neuroligins in cerebellar Purkinje cells have a role in the formation of climbing fiber synapses, but not parallel-fiber synapses (*Zhang et al., 2015*). Cadherin-9 is involved in the formation and differentiation of DGC–CA3 synapses, but not CA3–CA3 synapses (*Williams et al., 2011*). Neuroligin2 regulates inhibitory synaptic function in a pathway-specific manner (*Gibson et al., 2009*). Thus, it appears that there are input specific synaptic organizers that cooperate to establish precise networks in the brain. Our work on IGF2, which is specific to DGC–CA3 synapses, would add an example of how our brain utilizes specific molecules to emerge synapse specificity.

It will be also interesting to examine whether stabilization of inhibitory synapses is regulated by a similar feedback pathway. In CA3, FGF7 acts as a target-derived presynaptic organizing molecule for inhibitory synapses (*Terauchi et al., 2010*). Our preliminary screen suggests that FGF22 and FGF7 may regulate distinct molecules in excitatory (DGCs) vs. inhibitory neurons for synapse stabilization, because *Igf2* was not identified as a target gene of FGF7.

### Implication in neuropsychiatric disorders

IGF2 is implicated in behavioral phenotypes such as fear extinction (*Agis-Balboa et al., 2011*; *Agis-Balboa and Fischer, 2014*), depression (*Luo et al., 2015*), and memory consolidation and enhancement in rodents (*Chen et al., 2011*; *Pascual-Lucas et al., 2014*). Administration of IGF2 rescues spine formation and excitatory synaptic function in the hippocampus of a mouse model of Alzheimer's disease (*Pascual-Lucas et al., 2014*). In addition, *Igf2*$^{-/-}$ mice are resistant to acquiring epileptiform events in response to kainate administration (*Dikkes et al., 2007*). Interestingly, *Fgf22*$^{-/-}$ mice show seizure resistant phenotype (*Terauchi et al., 2010*) and depression-like behavior (*Williams et al., 2016*). Thus, defects in synapse stabilization through the FGF22–IGF2 pathway may be involved in diseases like epilepsy and depression. Our work may help develop new treatment strategies for such neuropsychiatric disorders.

## Materials and methods

### Animals

*Fgf22*$^{-/-}$ mice were described previously (*Terauchi et al., 2010*). The strain was maintained on the C57/BL6 background. *Fgf22*$^{flox/flox}$ mice (*Fgf22*$^{tm1a(EUCOMM)Hmgu}$) were from EUCOMM. *Grik4-Cre* mice were from Jackson (*Nakazawa et al., 2002*). *Igf2*$^{-/-}$ mice were described previously (*Lehtinen et al., 2011*). Both males and females of these knockout and littermate control mice were used in our study. C57/BL6 or ICR mice (Jackson Laboratory and Charles River Laboratories) were used to prepare cultures. All animal care and use was in accordance with the institutional guidelines and approved by the Institutional Animal Care and Use Committees at Boston Children's Hospital and University of Michigan.

## Microarray

DGCs were dissected from P14 WT and *Fgf22*[-/-] mice (n=4 per genotype). RNA was prepared with the RNeasy kit (Qiagen, Germantown, MD), and its quality was verified with Agilent Bioanalyzer using PICO chips. Microarray was performed on Illumina Bead station 500 with mouse-6 expression beadchip. *Igf2* was identified as one of the most down-regulated genes in *Fgf22*[-/-] DGCs (Diff Score = -43; see *Table 1*).

## In situ hybridization

In situ hybridization was performed as described (*Schaeren-Wiemers and Gerfin-Moser, 1993*). Digoxigenin-labeled cRNA probes were generated by in vitro transcription using DIG RNA labeling mix (Roche, Switzerland). The probe for *Igf2* was generated from the 3' untranslated region of the mouse *Igf2* cDNA (Open Biosystems, Lafayette, CO). In situ images were taken with a Nikon Coolpix 990 distal camera (Nikon, Japan) attached to an Olympus BX61 upright microscope (Olympus, Japan) under bright-field optics with 4x, 10x and 20x objective lenses.

## DNA constructs

The expression plasmid for IGF2 was generated by subcloning the full-length mouse *Igf2* cDNA into the NheI/XhoI sites of APtag-5 (GenHunter, Nashville, TN). The expression plasmid for IGF2-EGFP was generated by subcloning the full-length mouse *Igf2* cDNA (minus the stop codon) into the NheI/HindIII sites of pEGFP-N1 (Clontech, Mountain View, CA) in frame. The expression plasmid for Kir2.1 was generated by subcloning the full-length *Kir2.1* cDNA into SmaI/XhoI sites of pCMV-SPORT6. The IGF2 specific shRNA expression plasmids were constructed using synthetic oligonucleotides, which were cloned into the BamHI/HindIII sites of the HuSH shRNA vector (pRFP-C-RS, OriGene, Rockville, MD). The following shRNA targeting sequences were used: IGF2-shRNA#1, CGGACCGCGGCTTCTACTTCAGCAGGCCT and IGF2-shRNA#2, GTTGGTGCTTCTCATCTC TTTGGCCTTCG. The efficiency of shRNA-mediated knockdown of IGF2 was confirmed by co-transfecting each IGF2 shRNA plasmid with the IGF2-EGFP plasmid into cultured hippocampal neurons and measuring the total EGFP fluorescence intensity relative to control shRNA transfected neurons (*Figure 8—figure supplement 1*).

The synaptophysin-YFP plasmid was described previously (*Terauchi et al., 2010*; *Toth et al., 2013*; *Umemori et al., 2004*). The synaptophysin-mCherry plasmid was a kind gift from M. Sutton (University of Michigan).

## Primary neuronal cultures and transfection

Hippocampi were dissected from P0 mice, and hippocampal cells were dissociated in a solution containing 0.5% trypsin and 0.02% DNase I as described previously (*Terauchi et al., 2010*). $3–5 \times 10^4$ hippocampal cells were plated on a poly-D-lysine coated glass coverslip (diameter 12 mm, No.1, Carolina Biological, Burlington NC) and cultured in neurobasal media supplemented with B27 (Invitrogen, Waltham, MA). Transfection was performed using the CalPhos Mammalian transfection kit (Clontech). Cultured cells were transfected at 1–3DIV with 1.5–2.2 μg of plasmid DNA per coverslip. Recombinant FGF22 and IGF2 (both from R&D systems, Minneapolis, MN) were applied at 2 nM (FGF22) or 1.35 nM (IGF2) into culture medium at 1–3DIV. For neuronal activity blockade, TTX was applied at 1 μM into culture medium every fourth day stating at the time of transfection or factor application.

## Microfluidic cell culture

The chambers fabricated in polydimethylsiloxane (Xona Microfluidics SND450, Xona Microfluidics, Temecula, CA) were placed on a poly-D-lysine coated glass coverslip (25 x 25 mm, No.1, Carolina Biological) by physical contact. The chamber consists of two microfluidic compartments, somal and axonal sides, which are connected via microgrooves with a high fluidic resistance. Hippocampal cells (22,500 cells) were plated in the somal side compartment. Fluidic isolation of the axonal side compartment was established by applying 180 μl of media in the somal side and 110 μl in the axonal side. At 2DIV, 2.5 nM of FGF22 was applied into the axonal compartment. Cells were stained at 8DIV.

## Immunostaining and antibodies

Mouse brains were perfused with 4% paraformaldehyde (PFA) in PBS followed by further fixation in 4% PFA in PBS overnight. Sagittal and coronal sections were prepared on a cryostat (16–20 μm thick), and processed for staining. For IGF2 staining, sections were treated with methanol for 5 min at -20°C. For IGF2 staining together with calbindin or calretinin staining, sections were treated with acetone for 2 min at -20°C. Cultures were fixed with 3 or 4% PFA for 10 min at 37°C or methanol for 2–5 min at -20°C and stained as described previously (*Terauchi et al., 2010*). For immunostaining for IGF2, cultures were fixed with acetone for 2–3 min at -20°C. Dilutions and sources of antibodies are: monoclonal anti-calbindin (1:200; Sigma-Aldrich, St. Louis, MO; C9848), goat anti-calbindin (1:500; Frontier Institute, Japan), monoclonal anti-calretinin (1:200; Millipore, Billerica, MA; MAB1568), rabbit anti-calretinin (1:500; Synaptic Systems, Germany; 214102), monoclonal anti-Prox1 (1:1500; Millipore; MAB5652), rabbit anti-Prox1 (1:500; Millipore; AB5475), anti-VGLUT1 (1:5000; Millipore; AB5905), anti-VGAT (1:1500; Synaptic Systems; 131003), anti-PSD95 (1:700; Affinity Bioreagents, Golden, CO; MA-045 and 1:250; NeuroMab, Davis, CA; 75–028), anti-gephyrin (1:150; Synaptic Systems; 147021), anti-MAP2 (1:3000; Sigma-Aldrich; M4403), anti-neurofilament (1:1000; Covance, Princeton, NJ; SMI-312), rabbit anti-GFP (1:1000; Millipore; AB16901), chicken anti-GFP (1:2500; Aves Labs, Tigard, OR; GFP-1020), anti-DsRed (for staining of mCherry to enhance the fluorescence signal, 1:500; Clontech; 632496), anti-IGF2 (1:50 for brain section staining and 1:70 for cultured cell staining; Santa Cruz, Dallas, TX; sc-5622), anti-IGF2R (1:100; Santa Cruz; sc-25462), and antibody Py (1:50; a kind gift from M. Webb and P.L. Woodhams). When cells were co-stained with anti-Prox1 monoclonal antibody with another mouse IgG1 antibody, Zenon Alexa Fluor 568 Mouse IgG1 Labeling Kit (Invitrogen) was used to label anti-Prox1 monoclonal antibody.

## Electron microscopy

P28–P29 WT and *Igf2*[-/-] mice were perfused transcardially with fixative (2% PFA and 2.5% glutaraldehyde in 0.1 M cacodylate buffer pH 7.4), and their brains were postfixed for overnight. Hippocampi were removed, cross-sections (250 μm thick) were prepared, and small pieces (about 0.6 mm x 0.6 mm x 0.25 mm) of the SL and SR layers of the CA3 region were dissected. Dissected pieces were postfixed overnight, washed with 0.1 M cacodylate buffer pH 7.4, and treated with 2% osmium tetroxide in 0.1 M cacodylate buffer or 1.5% potassium ferrocyanide, 2% osmium tetroxide in 0.1 M cacodylate buffer for 1–5 hr. The samples were then rinsed with water, stained en bloc with 3% uranyl acetate for 1 hr, dehydrated in graded alcohols and propylene oxide, and embedded in TAAB 812 Resin (Canemco-Marivac, Canada). Blocks were kept for 48 hr at 60°C to complete polymerization. Both semi- and ultra-thin sections (10 and 70 nm) were prepared with Diatome Histo and Diatome Ultra 45° diamond knives, respectively, on Leica UC7 ultramicrotome (Leica, Germany), and observed with Tecnai G2 Spirit BioTWIN Transmission Electron Microscope. The digital images were captured with AMT 2k CCD camera system operated with AMT software (Advanced Microscopy Techniques Corp., Woburn, MA).

## Slice preparation and electrophysiology

For mEPSC recordings: Acute hippocampal slices were prepared from 5–7 month old mice. Mice were decapitated and the brains were removed, and 300 μm sections were cut using a Leica VT1000S vibratome. Sections were cut in an ice cold solution containing (in mM): 206 sucrose, 2.8 KCl, 2 MgSO$_4$, 1 MgCl$_2$, 1.25 NaH$_2$PO$_4$, 1 CaCl$_2$, 10 glucose, 26 NaHCO$_3$, and 0.4 ascorbic acid. Then, sections were incubated in an NMDG-HEPES recovery solution, containing (in mM): 92 NMDG, 92 HCl, 2.5 KCl, 10 MgSO$_4$, 0.5 CaCl$_2$, 1.2 NaH$_2$PO$_4$, 20 HEPES, 30 NaHCO$_3$, 25 glucose, 5 sodium ascorbate, 2 thiourea, and 3 sodium pyruvate, for 15 min at 34°C before putting the slices into artificial cerebral spinal fluid (aCSF) for 1 hr at room temperature. aCSF contained (in mM): 124 NaCl, 2.8 KCl, 2 MgSO$_4$, 1.25 NaH$_2$PO$_4$, 2 CaCl$_2$, 10 glucose, 26 NaHCO$_3$, and 0.4 ascorbic acid. All solutions were continuously bubbled with 95% O$_2$/5% CO$_2$. Neurons were visualized using a customized Scientifica/Olympus microscope. Data were obtained with a Multiclamp 700B amplifier (Axon Instruments, Union City, CA), digitized with Digidata 1440A (Axon Instruments) and collected with Clampex 10.0 (Axon Instruments). Whole-cell patch-clamp recordings were conducted with 4–6 MΩ pipette containing (in mM) 135 K-MeSO$_4$, 7 NaCl, 10 HEPES, 4 Mg-ATP, 0.3 Li-GTP, and 7 phosphocreatine. Cells were held at -70 mV. aCSF was supplemented during recording with 500 nM

tetrodotoxin and 50 µM picrotoxin and warmed to 32°C. mEPSCs were analyzed using Minianalysis (Synaptosoft, Decatur, GA).

For fEPSP recordings: mice (2–3 months old) were decapitated and the hippocampal lobules cut in the same solution as above. Transverse slices (400 µm) of the hippocampus were then cut using a tissue chopper (Stoelting, Kiel, WI). After slicing, sections were incubated in an NMDG-HEPES recovery solution for 15 min at 34°C before putting the slices into aCSF for 1 hr. Slices were then incubated in aCSF at room temperature for at least 1 hr before recording. Then, slices were transferred to a recording chamber, maintained at 32°C and continuously perfused at 1–2 ml/min with oxygenated aCSF. Recording electrodes were pulled from borosilicate capillary glass and filled with 1 M NaCl, 25 mM HEPES (1.5 mm o.d.; Sutter Instruments, Novato, CA). The recording pipette was placed in the in the SL or SR layers of the CA3 region of the hippocampus. Recordings were made with a MultiClamp 700B amplifier, collected using Clampex 10.3, and analyzed using Clampfit 10.3. fEPSPs were evoked using cluster electrodes (FHC) placed in the SL or SR layers of the CA3 region of the hippocampus. Current between 0.1–1 mA for 0.1 s was used to elicit a response. Maximum responses were then used for paired-pulse facilitation experiments. 1 µM DGC-IV was then added to the aCSF and the experiments were repeated after 10 min of perfusion with drug.

## Imaging

Fluorescent images were taken on epi-fluorescence microscopes (Olympus BX61 and BX63) or confocal microscopes (Olympus FV1000 and Carl Zeiss LSM700, Germany). With epi-fluorescence microscopes, 12-bit images at a 1,376 x 1,032 (Olympus BX61) or 1,376 x 1,038 (Olympus BX63) pixel resolution were acquired with 40x and 20x objective lenses using an F-View II CCD camera (Soft Imaging System, Germany) or an XM10 Monochrome camera (Olympus). With confocal microscopes, 12-bit (FV1000) or 8-bit (LSM700) images at a 1,024 x 1,024 pixel resolution were obtained using 20x, 40x and 63x objective lenses with a 1.0 or 1.5x zoom. Images were acquired as a z-stack (17–20 optical sections, 0.4 µm step size). Images in the same set of experiments were acquired with the identical acquisition settings regarding the exposure time, laser power, detector gain, or amplifier offset.

The intensity of stained signals and the size and density of stained puncta were quantified and analyzed using MetaMorph software. For images of hippocampal sections, the staining intensity in the lateral ventricle was calculated as the background signal and subtracted from each image. For images of cultured neurons stained for synaptic proteins, the staining intensity of the dendritic shaft in control cultures was calculated as the background signal and subtracted from each image. For images of cultured neurons stained for anti-IGF2 antibody, the lowest staining intensity in the culture was calculated as the background signal and subtracted from each image. For colocalization analyses, objects were considered to colocalize if more than 25% of the object was overlapped with the other object.

## Statistical analysis

The statistical tests performed were two-tailed Student's *t*-test or Two-way ANOVA, as indicated in the figure legend. Two-way ANOVA was followed by Tukey's *post hoc* test. All data are expressed as mean ± s.e.m. No statistical methods were used to pre-determine sample sizes, but our sample sizes were similar to those reported in previous publications in the field (*Murata and Constantine-Paton, 2013*; *Sharma et al., 2010*; *Terauchi et al., 2010*; *Toth et al., 2013*). No data points were excluded from any experiments.

## Acknowledgements

We thank Mei Zhang, Patricia Lee and Hillary Mullan for animal care; Aaron Reifler and Robert Thompson for help with microarray screen; Gorski Grzegorz for help with electron microscopy; Ania Dabrowski and Erin Piell for technical assistance; Masahiro Yasuda and Sivapratha Nagappan Chettiar for critical reading of the manuscript. This work was supported by NIH grants R01-NS070005 and R01-MH091429 (HU) and R00-NS072192 (ML) as well as by the Bipolar Disorder Fund for Neuroscience Research at Harvard University, supported by Kent and Liz Dauten (HU).

## Additional information

### Funding

| Funder | Grant reference number | Author |
| --- | --- | --- |
| National Institute of Neurological Disorders and Stroke | R01-NS070005 | Hisashi Umemori |
| National Institute of Mental Health | R01-MH091429 | Hisashi Umemori |
| National Institute of Neurological Disorders and Stroke | R00-NS072192 | Maria Lehtinen |
| Bipolar Disorder Fund for Neuroscience Research at Harvard University, supported by Kent and Liz Dauten | | Hisashi Umemori |

The funders had no role in study design, data collection and interpretation, or the decision to submit the work for publication.

### Author contributions

AT, Designed the experiments, Wrote the manuscript, Performed the experiments, Analysis and interpretation of data; EMJ-V, Performed the experiments, Analysis and interpretation of data, Drafting or revising the article; BB, Performed the experiments, Analysis and interpretation of data; MKL, Provided critical materials, Drafting or revising the article, Contributed unpublished essential data or reagents; HU, Designed the experiments, Wrote the manuscript, Secured funding and supervised the project, Acquisition of data, Analysis and interpretation of data

### Author ORCIDs

Hisashi Umemori, http://orcid.org/0000-0001-7198-2062

### Ethics

Animal experimentation: All animal care and use in this study was in accordance with the institutional guidelines and approved by the Institutional Animal Care and Use Committees at Boston Children's Hospital (#13-11-2528) and University of Michigan (#PRO00003549).

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
