## [Decision Letter]

Thank you for submitting your work entitled "Retrograde FGF22 signaling regulates IGF2 expression for activity-dependent synapse stabilization in the mammalian brain" for consideration by *eLife*. Your article has been favorably evaluated by Gary Westbrook (Senior editor) and three reviewers, one of whom is a member of our Board of Reviewing Editors. One of the three reviewers has agreed to reveal his identity: Jaewon Ko.

The reviewers have discussed the reviews with one another and the Reviewing Editor has drafted this decision to help you prepare a revised submission.

Editor/reviewer summary:

In the rodent hippocampus, synapse formation starts in the first postnatal week, followed by activity-dependent maturation and stabilization stages. The authors previously showed that target-derived FGF22 promoted presynaptic differentiation of excitatory synapses between dentate gyrus granule cells (DGCs) and CA3 pyramidal neurons in the hippocampus (Terauchi et al., Nature 2010). They also showed that another target-derived factor, SIRPα, regulated synapse maturation of DGCs during postnatal weeks 2-4 (Toth et al., Nat Neurosci 2013). In this manuscript, the authors report decreased IGF2 expression in calretinin-positive DGCs in total and CA3-specific FGF22-null mice in vivo, and that bath and local application of FGF22 induced expression of IGF2 specifically in calretinin-positive immature DGCs in vitro. Overexpressed IGF2-EGFP localized to DGC axons in an activity-dependent manner and promoted presynaptic differentiation, as measured by accumulation of synaptophysin-YFP. In addition, treatment with IGF2 restored excitatory presynapse differentiation in FGF22-null CA3 neurons in vitro without changes in postsynaptic markers. FGF22-dependent accumulation of synaptophysin-YFP in cultured wild-type DGCs required IGF2 only after 6DIV. Finally, excitatory synapses were reduced in IGF2-null calretinin-positive DGCs in vivo. From these data, authors conclude that target-derived FGF2 induces expression of IGF2, which is critical for the activity-dependent stabilization of DGC presynaptic terminals on CA3 neurons. It has not been clear whether and how target-derived molecules control gene transcription and stabilization of presynaptic terminals in the mammalian CNS. This study that sheds light on the novel role of FGF22-induced IGF2 in activity-dependent stabilization of presynaptic terminals in the hippocampus. However, several key issues need to be addressed to support the authors' claim.

Essential revisions:

1) Most of the data presented in this manuscript are of high quality and convincing, except Figure 3. Immunoreactive signals from IGF2 antibody are too faint and weak. Because the authors have potent IGF2 knockdown vector (Figure 8) and IGF2 knockout mice (Figure 9), the authors could validate the IGF2 antibody using these reagents.

2) The decrease in the frequency of mEPSCs in IGF2 knockout mice (Figure 9) could be explained by a decrease in the release sites (inputs) and/or a decrease in the probability of neurotransmitter release. The authors could analyze the paired-pulse ratio to help interpret the data.

3) It is not completely clear at which stage of synapse development IGF2 works. As the same authors classified previously (Toth et al., Nat Neurosci 2013), three different stages of synapse development can be defined for DGC-CA3 synapses: P0-P14, initial synapse differentiation; P15-P29, synapse maturation; P30-P44, synapse maintenance. In most experiments, authors examined the effect of IGF2 at 10DIV (Figure 4–Figure 6) or 13DIV (Figure 7). Indeed, IGF2 is highly expressed in calretinin-positive DGCs, which mostly correspond to neurons born 3-17.5 days ago (Figure 1). Thus, IGF2 appears to be a molecule that regulates initial synapse "differentiation" after FGF22. The authors claim that IGF2 is dispensable for initial presynaptic differentiation, but critical for presynaptic stabilization based on Figure 8. However, in this experiment, FGF22 was applied to hippocampal neurons transfected with shRNA against IGF2 at 1DIV and examined at 6DIV or 12DIV. FGF22-induced synaptophysin accumulation was observed in neurons fixed at 6DIV, but not those at 12DIV. However, prolonged knockdown of IGF2 could have non-specific effects on neurons, such as reduction of synaptic vesicles and reduced expression of FGF receptors. To examine the role of IGF2 at later synapse developmental stages, the authors need to knockdown IGF2 at later time points.

4) It is interesting that FGF22-induced IGF2 expression only in calretinin-positive young DGCs (Figure 3), but not in other DGCs, which also respond to FGF22. However, IGF2 immunoreactivity remained unaffected in calbindin-positive DGCs in FGF22-null mice (Figure 1), indicating that IGF2 can be expressed in an FGF-independent manner. Many questions remain unanswered here. What determines the specific responsiveness of calretinin-positive DGCs to FGF22? How is IGF2 expressed in calbindin-positive DGCs in FGF22-null mice? What roles does IGF2 play in calbindin-positive DGCs? Why do DGCs express IGF2 by two different pathways (i.e., FGF22-dependent and independent) at different developmental stages? The authors need to provide some explanations. In most of the vitro experiments, the authors examined the roles of IGF2 on presynaptic differentiation in prox1-positive DGCs, which includes many calretinin-negative DGCs. Considering that FGF22-induced IGF2 expression was specifically observed in calretinin-positive DGCs, the authors need to focus on these cells, or at least show what percentage of their cultured neurons corresponds to calretinin-positive DGCs.

5) The model shown in Figure 9 is attractive, but is not completely supported by their data. First, it is unclear where and how IGF2 is located on presynaptic sites. Is overexpressed IGF2-EGFP localized on the surface of presynaptic terminals (Figure 4)? If so, depending on where and how IGF2 is released, IGF2 should not be considered as a local, but a global, presynaptic organizer. Second, it is unclear where and how neuronal activity is required for the effect of IGF2 (Figure 6). If IGF2 is secreted by neuronal activity as the authors suggest in the Discussion, why didn’t tetrodotoxin (TTX) treatment increase intracellular IGF2 levels? IGF2 may be transported to presynaptic terminals by neuronal activity, but why didn’t TTX treatment affect transport of synaptic vesicles (Figure 6)? Neuronal activity may affect the exocytosis/endocytosis ratio or stabilization of IGF2 in axons. It is also possible that TTX treatment for 7 days from 1DIV may change expression of various genes that affect axon branching and expression of IGF2 receptors. Changes in postsynaptic neuronal activity may also contribute to presynaptic accumulation of synaptophsyin-YFP by some other mechanisms. These issues require further clarification and discussion.

6) The role of endogenous IGF2 at DGC-CA3 synapses in vivo is not completely clear from the data shown in Figure 9. Because the authors recorded from CA3 neurons, mEPSCs originating from synapses at the strata radiatum and lucidum are mixed. The authors need to show that mossy fiber-evoked EPSC(P)s are specifically affected in IGF2-null mice. In addition, changes in presynaptic differentiation should be assessed by paired-pulse stimulation of specific input fibers. Electron microscopic analyses of pre- and postsynaptic structures of DGC-CA3 synapses in IGF2-null mice would also strengthen the authors' argument.

---

## [Author Response]

Essential revisions:

*1) Most of the data presented in this manuscript are of high quality and convincing, except Figure 3. Immunoreactive signals from IGF2 antibody are too faint and weak. Because the authors have potent IGF2 knockdown vector (Figure 8) and IGF2 knockout mice (Figure 9), the authors could validate the IGF2 antibody using these reagents.* We appreciate the comment. As suggested, we have validated the IGF2 antibody using IGF2 knockout (KO) mice. As shown in Figure 1—figure supplement 2, the antibody did not stain IGF2KO sections both in the dentate granule cells (DGCs) and choroid plexus, where IGF2 is known to be highly expressed (Lun et al., 2015; Stylianopoulou et al., 1988; Ayer-le Lievre et al., 1991). Please note that in Figure 3, IGF2 signals are high in cultured DGCs treated with FGF22, but not in DGCs without FGF22 (Figure 3) or in CA3 neurons (Figure 3).

*2) The decrease in the frequency of mEPSCs in IGF2 knockout mice (Figure 9) could be explained by a decrease in the release sites (inputs) and/or a decrease in the probability of neurotransmitter release. The authors could analyze the paired-pulse ratio to help interpret the data.* We appreciate the comment. To further characterize the synaptic defects in IGF2KO mice, we have performed additional electrophysiological recordings and electron microscopic analyses.

Electrophysiological analysis for DGC-CA3 connections revealed that i) the maximum fEPSP slope is significantly lower in IGF2KO than WT controls and ii) the paired pulse ratio is significantly lower in IGF2KO than WT controls.

Electron microscopic analysis of synapses in the CA3 stratum lucidum layer revealed that i) there are similar numbers of synapses as assessed by the number of postsynaptic densities between WT and IGF2KO mice and ii) there are fewer synaptic vesicles, less clustering of synaptic vesicles, and fewer docked vesicles in the presynaptic terminals in IGF2KO relative to WT mice.

These results suggest that excitatory synaptic transmission at DGC-CA3 synapses is impaired in IGF2KO mice due to the loss of synaptic vesicles from the presynaptic terminals. These results are included in Figure 10 and Figure 11, and described in the manuscript.

*3) It is not completely clear at which stage of synapse development IGF2 works. As the same authors classified previously (Toth et al., Nat Neurosci 2013), three different stages of synapse development can be defined for DGC-CA3 synapses: P0-P14, initial synapse differentiation; P15-P29, synapse maturation; P30-P44, synapse maintenance. In most experiments, authors examined the effect of IGF2 at 10DIV (Figure 4–Figure 6) or 13DIV (Figure 7). Indeed, IGF2 is highly expressed in calretinin-positive DGCs, which mostly correspond to neurons born 3-17.5 days ago (Figure 1). Thus, IGF2 appears to be a molecule that regulates initial synapse "differentiation" after FGF22. The authors claim that IGF2 is dispensable for initial presynaptic differentiation, but critical for presynaptic stabilization based on Figure 8. However, in this experiment, FGF22 was applied to hippocampal neurons transfected with shRNA against IGF2 at 1DIV and examined at 6DIV or 12DIV. FGF22-induced synaptophysin accumulation was observed in neurons fixed at 6DIV, but not those at 12DIV. However, prolonged knockdown of IGF2 could have non-specific effects on neurons, such as reduction of synaptic vesicles and reduced expression of FGF receptors. To examine the role of IGF2 at later synapse developmental stages, the authors need to knockdown IGF2 at later time points.* We appreciate the comment. In our previous papers, we have defined the stages of synapse development based on its dependence on neural activity: in the hippocampus, synapse development from P0 to ~P14 is not apparently influenced by neural activity ("initial synapse differentiation"), but that from ~P14 to ~P28 is regulated by activity, where we found that activity-dependent synapse maturation (e.g., mediated by SIRP; Toth et al., 2013) and activity-dependent synapse elimination (Yasuda et al., 2011) take place. Because the effects of IGF2 are dependent on neural activity, we proposed that IGF2 is critical for "synapse maturation/stabilization" in the manuscript. As suggested, we have further examined the role of IGF2 at later stages of synapse development both in vitro and in vivo.

In vitro, we knocked down IGF2 at a later synapse developmental stage, from DIV6, and examined the effects at DIV12. We found that IGF2 knockdown from DIV6 decreased the number and size of presynaptic terminals (as assessed by synaptophysin-YFP) and blocked the synaptogenic effects of FGF22. Since IGF2 knockdown from DIV1-DIV6 did not affect presynaptic terminals or synaptogenic effects of FGF22 (Figure 8), our result suggests that IGF2 is indeed critical for a later stage of synapse development (after DIV6). This new result is included in Figure 8.

In vivo, we analyzed the time course of presynaptic development in IGF2KO mice from P8 to P29. We found that in the SL layer of CA3, IGF2KO mice show no presynaptic defects at P8, but start to show mild defects at P14 (the number of VGLUT1 puncta was normal, but their size was decreased), and exhibit significant defects at P21 and P29 (both the number and size of VGLUT1 puncta were significantly decreased). We did not detect any changes in the SR layer of CA3. Thus, IGF2 appears to be mainly important for DGC-CA3 synapse development after P14. These results are included in Figure 9.

These results suggest that IGF2 is not critical for initial synapse development, but is important for the later stages of synapse development (after DIV6 or P14). As IGF2 effects are activity dependent (Figure 6), we propose that IGF2 is mainly important for the stages of synapse maturation and stabilization. These results are also discussed in the text (subsection “Temporal and Spatial Specificity of IGF2 Effects”, first paragraph).

4) It is interesting that FGF22-induced IGF2 expression only in calretinin-positive young DGCs (Figure 3), but not in other DGCs, which also respond to FGF22. However, IGF2 immunoreactivity remained unaffected in calbindin-positive DGCs in FGF22-null mice (Figure 1), indicating that IGF2 can be expressed in an FGF-independent manner. Many questions remain unanswered here. What determines the specific responsiveness of calretinin-positive DGCs to FGF22? How is IGF2 expressed in calbindin-positive DGCs in FGF22-null mice? What roles does IGF2 play in calbindin-positive DGCs? Why do DGCs express IGF2 by two different pathways (i.e., FGF22-dependent and independent) at different developmental stages? The authors need to provide some explanations.

We agree with the reviewer that because IGF2 is still expressed in calbindin-positive DGCs in *Fgf22^-/-^*mice, IGF2 can be expressed in an FGF22-independent manner as well. Indeed, IκB, which is not utilized by FGF22 signaling, has been shown to induce expression of IGF2 for spine maturation in mature hippocampal neurons (Schmeisser MJ, et al., 2012). The expression of the mouse *Igf2* gene is regulated by three alternative promoters (Sasaki et al., 1992). Thus, different signals seem to be used to regulate distinct phases of neuronal development.

Specific responsiveness of calretinin-positive DGCs to FGF22 appears to be linked to the developmental stage. Calretinin-positive DGCs elongate axons to CA3 and contact CA3 pyramidal neurons to form synapses (Ming and Song, 2005; Li et al., 2009; Aguilar-Arredondo A et al., 2015; Yasuda et al., 2011), and that is when DGCs respond to FGF22. Thus, around that stage, DGCs may become more responsive to FGF22. We have included these points in the Discussion (subsection “Temporal and Spatial Specificity of IGF2 Effects”, first paragraph).

In most of the vitro experiments, the authors examined the roles of IGF2 on presynaptic differentiation in prox1-positive DGCs, which includes many calretinin-negative DGCs. Considering that FGF22-induced IGF2 expression was specifically observed in calretinin-positive DGCs, the authors need to focus on these cells, or at least show what percentage of their cultured neurons corresponds to calretinin-positive DGCs.

We appreciate the comment. As suggested, we examined the percentage of calretinin-positive DGCs in culture by co-staining for calretinin and prox1. We found that a majority of DGCs (around two thirds) were calretinin positive in our culture. Thus, our results with DGCs mostly reflect results from calretinin-positive DGCs. We have added these data in Figure 3—figure supplement 1 and described the results in the text.

5) The model shown in Figure 9 is attractive, but is not completely supported by their data. First, it is unclear where and how IGF2 is located on presynaptic sites. Is overexpressed IGF2-EGFP localized on the surface of presynaptic terminals (Figure 4)? If so, depending on where and how IGF2 is released, IGF2 should not be considered as a local, but a global, presynaptic organizer.

We examined whether IGF2-EGFP is localized on the surface of neurons by staining transfected neurons without a detergent (with anti-GFP followed by Alexa Fluor 647). We found that ~40% of IGF2-EGFP is localized on the surface of neurons. Furthermore, surface IGF2-EGFP is always localized at presynaptic terminals, as it shows perfect colocalization with synaptophysin-mCherry. A majority of intracellular IGF2 is also localized at presynaptic terminals (~75%).

In addition, we found that the major receptor for IGF2, IGF2R, is also localized at presynaptic terminals (a minor receptor for IGF2, IGFR1, is also localized at synapses; Gazit et al., 2016).

These results are consistent with the notion that IGF2 is secreted from the presynaptic terminal and binds to IGF2R, which is also localized at the presynaptic terminal, so that IGF2 acts locally in an autocrine manner for its effects. These data are included in Figure 4 and Figure 4—figure supplement 1.

We do agree that it is possible that secreted IGF2 may also act as a global presynaptic organizer if IGF2 is released from the presynaptic terminal. We have discussed this possibility in the text (subsection “A Novel Feedback Pathway through Transcriptional Regulation for Presynaptic Stabilization in the Mammalian Brain”, second paragraph).

Second, it is unclear where and how neuronal activity is required for the effect of IGF2 (Figure 6). If IGF2 is secreted by neuronal activity as the authors suggest in the Discussion, why didn’t tetrodotoxin (TTX) treatment increase intracellular IGF2 level?. IGF2 may be transported to presynaptic terminals by neuronal activity, but why didn’t TTX treatment affect transport of synaptic vesicles (Figure 6)? Neuronal activity may affect the exocytosis/endocytosis ratio or stabilization of IGF2 in axons. It is also possible that TTX treatment for 7 days from 1DIV may change expression of various genes that affect axon branching and expression of IGF2 receptors. Changes in postsynaptic neuronal activity may also contribute to presynaptic accumulation of synaptophsyin-YFP by some other mechanisms. These issues require further clarification and discussion.

We appreciate the comments. To further address the role of activity on IGF2 localization and function, we suppressed intrinsic neuronal excitability by overexpressing the inwardly rectifying potassium channel Kir2.1. We found that co-expression of Kir2.1 with IGF2 decreased synaptic localization of IGF2 as well as its synaptogenic function, similarly to the effects of TTX. These experiments suggest that intrinsic neuronal excitability of the presynaptic neurons is critical for IGF2 localization and thus, for its synaptogenic function.

We think that since neural activity is important for IGF2 transportation, TTX/Kir2.1 did not increase the intracellular IGF2 levels. We also think that IGF2's effects are for the stabilization of synaptic vesicles but not for the initial recruitment of synaptic vesicles (which is carried out by FGF22), and thus, TTX/Kir2.1 did not affect synaptic vesicle transportation. We agree that neural activity may affect the secretion of IGF2, and we plan to utilize IGF2-pHluorin to test this idea in future studies. We also agree that various activity-dependent genes might influence IGF2/IGF2R localization, which is an interesting future study. Our Kir2.1 experiments, in which Kir2.1 was sparsely transfected so that we can ignore effects from postsynaptic neurons, suggest that changes in postsynaptic activity do not play critical roles in IGF2 localization and function. We have included the data with Kir2.1 in Figure 6 and discussed these points in the text (subsection “Activity-Dependent Regulation of Presynaptic Stabilization by IGF2”).

*6) The role of endogenous IGF2 at DGC-CA3 synapses* in vivo *is not completely clear from the data shown in Figure 9. Because the authors recorded from CA3 neurons, mEPSCs originating from synapses at the strata radiatum and lucidum are mixed. The authors need to show that mossy fiber-evoked EPSC(P)s are specifically affected in IGF2-null mice. In addition, changes in presynaptic differentiation should be assessed by paired-pulse stimulation of specific input fibers. Electron microscopic analyses of pre- and postsynaptic structures of DGC-CA3 synapses in IGF2-null mice would also strengthen the authors' argument.*

We appreciate the comment. To further characterize the synaptic defects in IGF2KO mice, we have performed additional electrophysiological recordings and electron microscopic analyses.

Electrophysiological analysis revealed that i) the maximum fEPSP slope for DGC-CA3 connections, but not for CA3-CA3 connections, is significantly lower in IGF2KO than WT controls and ii) the paired pulse ratio for DGC-CA3 connections, but not for CA3-CA3, is significantly lower in IGF2KO than WT controls.

Electron microscopic analysis revealed that i) there are similar numbers of synapses in the stratum lucidum (SL: where DGC-CA3 synapses locate) and stratum radiatum (SR: where CA3-CA3 synapses locate) layers as assessed by the number of postsynaptic densities between WT and IGF2KO mice and ii) there are fewer synaptic vesicles, less clustering of synaptic vesicles, and fewer docked vesicles in the presynaptic terminals in the SL layer, and not the SR layer, of IGF2KO relative to WT mice.

These results suggest that excitatory synaptic transmission at DGC-CA3, and not CA3-CA3, synapses is specifically impaired in IGF2KO mice due to the loss of synaptic vesicles from the presynaptic terminals in the SL, and not SR, layer. Thus, our results show that IGF2 is critical for the stabilization/maintenance of synaptic vesicles specifically at the DGC-CA3 synapses. These results are included in Figure 10 and Figure 11, and described in the manuscript.